# Structured Noise Adaptation for Sequential Bayesian Filtering with Embedded Latent Transfer Operators

**Naichang Ke**                                                    *naichang.ke@ist.osaka-u.ac.jp*
*The University of Osaka*

**Pongpisit Thanasutives**                                         *pongpisit.thanasutives@riken.jp*
*RIKEN Center for Advanced Intelligence Project (AIP)*

**Yoshinobu Kawahara**                                             *kawahara@ist.osaka-u.ac.jp*
*The University of Osaka & RIKEN Center for Advanced Intelligence Project (AIP)*

**Reviewed on OpenReview:** *https://openreview.net/forum?id=smFAyzvh5r*

## Abstract

Kalman filters based on the Embedded Latent Transfer Operators (ELTO) emerge as novel statistical tools for sequential state estimation. However, a critical limitation stems from their use of simplified noise models, which fail to dynamically adapt to non-stationary processes. To address this limitation, we introduce an ELTO-based Bayesian filtering approach with a new structured parameterization for the filter's noise model. This parameterization enables structured noise adaptation, which couples the data-driven learning of an optimal time-invariant noise model with dynamic parameter adaptation that responds to changes in dynamics within non-stationary processes. Empirical results show that our structured noise adaptation improves the filter's dynamic state estimation performance in noisy, time-varying environments.

## 1 Introduction

Sequential state estimation is the process of continuously updating an estimate of a system's state over time based on incoming observations. It serves as a foundation for many real-world applications, including robotics, navigation, and financial modeling (Gebhardt et al., 2019; Greenberg et al., 2023; DeMiguel et al., 2024). A widely used approach to state estimation is Kalman filtering (Kalman, 1960), which performs sequential Bayesian state estimation by iteratively propagating a latent state and its uncertainty forward in time (Fukumizu et al., 2013), and then updating on new observations. Achieving high state estimation performance is, however, often hindered by a fundamental problem: specifying a noise model that remains robust under the non-stationarity of real-world processes (Masreliez & Martin, 2003). A common approach to this problem is, for example, down-weighting measurement outliers at the update step, which merely treats the symptom of corrupted data (Duran-Martin et al., 2024; Wang et al., 2018; Agamennoni et al., 2012). In contrast, we address the problem by improving the robustness of the filter's noise model in learning the underlying system dynamics from noisy data.

Our proposed method builds on the work of (Gebhardt et al., 2019), which introduces a computationally efficient state estimation technique called the kernel Kalman rule (KKR). Similar to the kernel Bayes' filter (KBR) (Fukumizu et al., 2013), KKR, when combined with the kernel sum rule, uses conditional embedding operators (e.g., the transfer operator) to formulate kernel Kalman filtering (KKF) in the reproducing kernel Hilbert space (RKHS). By performing state estimation in the RKHS, KKF overcomes the explicit parametric assumptions required in the original state space.

Given high-dimensional observations, (Ke et al., 2025) have recently proposed a spectral learning algorithm, derived from stochastic realization theory (Katayama, 2005), to approximate data-driven representation of

the transfer operator governing the evolution of the embedded latent state in an RKHS. This resulting transfer operator is also known as the Embedded Latent Transfer Operator (ELTO). Because the spectral learning algorithm enables data-driven modeling of nonlinear stochastic processes using the ELTO, sequential state estimation (ELTO-KF) can directly combine these identified operators with the Bayesian inference procedure of the KKR. Nonetheless, the ELTO-KF approach inherits a critical limitation: the reliance on fixed noise covariances that fail to adapt to non-stationary processes.

Optimizing the noise models, i.e., the covariance matrices of process and measurement noise, of Kalman filters to ensure robustness against changes in the dynamics of non-stationary processes is an important challenge, because it is known that suboptimal noise covariance matrices severely degrade filtering performance (Greenberg et al., 2023). One noteworthy model-based solution is the adaptive extended Kalman filter (AEKF), which dynamically tunes the noise matrices using the filter's residual and innovation (Wang, 1999; Akhlaghi et al., 2017). Another class of solutions is data-driven, which treats the parameters of the noise models as learnable and uses numerical optimization to obtain the best time-invariant parameters in terms of fitting performance (Greenberg et al., 2023; Becker et al., 2019). Both strategies highlight the important role of the noise model. However, to facilitate the learning of complex system dynamics in high-dimensional contexts, a research gap remains in constructing a unified approach that benefits from both parameter learning strategies to ultimately improve the filtering performance.

In this paper, we improve the ELTO-KF by incorporating a new structured parameterization for the filter's noise model, proposing a novel ELTO-based adaptive Kalman filtering method, called ELTO-AKF. This parameterization enables structured noise adaptation, coupling the data-driven learning of the filter's optimal time-invariant noise model with dynamic parameter adaptation to enhance dynamic state estimation performance in non-stationary processes. Our proposed method incorporates adaptive estimation of the noise covariance matrices into the data-driven optimization using the filter's residual and innovation. This integration provides robust covariance representations for noisy non-stationary processes. Our structured parameterization of the noise covariance matrices reduces computational complexity compared to full-rank matrices, providing a tractable structure for dynamic parameter adaptation. The proposed data-driven approach enables the structured noise adaptation, which integrates data-driven parameter optimization with dynamic parameter adaptation, thereby addressing the aforementioned research gap. We demonstrate that ELTO-AKF is effective across a broad range of challenging scenarios, including non-stationary LiDAR trajectories, high-dimensional Lorenz systems, and a downstream denoising task for the sparse identification of partial differential equations (PDEs).

Our contributions are summarized as follows:

- We introduce a new structured parameterization for tractable representation learning of noise covariance matrices and propose the ELTO-AKF method, in which the parameterization is implemented.

- We demonstrate that structured noise adaptation improves the filter's dynamic state estimation performance in non-stationary processes by coupling the learning of a robust, time-invariant noise model via derivative-free optimization with dynamic parameter adaptation.

- We demonstrate the practical utility of ELTO-AKF as a robust filter for denoising scientific data in non-stationary processes. By effectively reducing noise in the observed states, our method improves the accuracy of downstream data-driven equation discovery, such as identifying governing PDEs.

## 2 Background

### 2.1 State Space Models

The state space model (SSM) is a mathematical framework for describing a dynamical system, consisting of a latent state process $\{\mathbf{x}(t) \in \mathbb{X}\}$ and an observation process $\{\mathbf{y}(t) \in \mathbb{Y}\}$. The system evolution is governed by a state transition density $p_{tr}$, which is defined for any measurable set $\mathbb{A}$ by $Pr(\mathbf{x}(t+1) \in \mathbb{A}|\mathbf{x}(t) = \boldsymbol{x}) = \int_{\mathbb{A}} p_{tr}(\boldsymbol{z}|\boldsymbol{x})d\boldsymbol{z}$, and an observation density $p_{ob}$, which links the latent state to the observations/measurements, defined by $Pr(\mathbf{y}(t) \in \mathbb{A}'|\mathbf{x}(t) = \boldsymbol{x}) = \int_{\mathbb{A}'} p_{ob}(\boldsymbol{y}|\boldsymbol{x})d\boldsymbol{y}$. Note that $\mathbb{A}'$ is another measurable set.

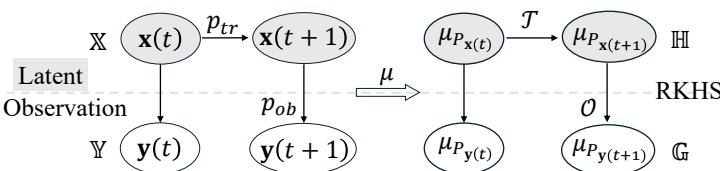

Figure 1: State-space representation of original and latent variables. $\mathcal{T}$ and $\mathcal{O}$ respectively represent the transfer and observable operators for the latent state embedded in the RKHS by $\mu$.

**SSMs in Reproducing Kernel Hilbert Spaces.** Since Hilbert-space distribution embeddings allow us to represent arbitrary probability distributions nonparametrically and perform inference entirely in this space, we embed the state densities into RKHSs. Given a measurable positive definite kernel $k$ on $\mathbb{X}$ (e.g., $\sup_{\boldsymbol{x} \in \mathbb{X}} k(\boldsymbol{x}, \boldsymbol{x}') < \infty$) and its corresponding feature map $\psi$, the probability density of the state $\mathbf{x}(t)$ is represented by a kernel mean embedding $\mu_{P_{\mathbf{x}(t)}}$ into the RKHS, denoted as $\mathbb{H}$. This embedding is defined as the mapping $\mu : \mathbb{M}_+(\mathbb{X}) \to \mathbb{H}, P \mapsto \int \psi(\boldsymbol{x}) dP(\boldsymbol{x})$ on measure space $\mathbb{M}_+(\mathbb{X})$ for any probability measure $P$ on $\mathbb{X}$. Within this embedded Hilbert space, the abstract transition and observation can be described by covariance operators. Let $(\boldsymbol{x}, \boldsymbol{y})$ be a random variable taking values in $\mathbb{X} \times \mathbb{Y}$ with a marginal distribution $P_{\mathbf{x}}$ and a joint distribution $P_{\mathbf{xy}}$, and let $\mathbb{H}$ and $\mathbb{G}$ be their corresponding RKHSs with feature maps $\psi$ and $\phi$. Following (Baker, 1973), the covariance operator $\mathcal{C}_{\mathbf{x}} : \mathbb{H} \to \mathbb{H}$ and cross-covariance operator $\mathcal{C}_{\mathbf{yx}} : \mathbb{H} \to \mathbb{G}$ are defined as:

$$\mathcal{C}_{\mathbf{x}} := \int \psi(\boldsymbol{x}) \otimes \psi(\boldsymbol{x}) dP_{\mathbf{x}}(\boldsymbol{x}),$$

$$\mathcal{C}_{\mathbf{yx}} := \int \phi(\boldsymbol{y}) \otimes \psi(\boldsymbol{x}) dP_{\mathbf{xy}}(\boldsymbol{x}, \boldsymbol{y}). \tag{1}$$

The conditional distribution is represented by the conditional embedding operator $\mathcal{C}_{\mathbf{y}|\mathbf{x}} := \mathcal{C}_{\mathbf{yx}} \mathcal{C}_{\mathbf{x}}^{-1}$ (Song et al., 2009).

**Data-driven Identification of System Operators via Spectral Learning.** (Ke et al., 2025) proposed a spectral learning algorithm to estimate matrix representations of system operators, namely the Embedded Latent Transfer Operator (ELTO) and the Embedded Observable Operator (EOO), which govern the evolution of the embedded latent state in the RKHS as well as its relationship to the observation. The latent state process $\{\mathbf{x}(t)\}$ is constructed in a data-driven way based on stochastic realization theory (Katayama, 2005). Consequently, the spectral learning enables data-driven identification of the system operators, $\mathcal{T} := \mathcal{C}_{\mathbf{x}(t+1)|\mathbf{x}(t)}$ and $\mathcal{O} := \mathcal{C}_{\mathbf{y}(t)|\mathbf{x}(t)}$, from the observation process $\{\mathbf{y}(t)\}$. It is shown that a data-driven approximation of system operators is more flexible and can be used to improve the performance of traditional model-based (adaptive) Kalman filters[1]. Both operators ($\mathcal{T}$ and $\mathcal{O}$ in Figure 1) are necessary for the kernel Kalman filtering described next.

## 2.2 Kernel Kalman Filtering

Sequential state estimation, i.e., Kalman filtering, is the primary task for the state-space models. The fundamental goal is to sequentially infer the full probability distribution of the latent state from a history of observations. Traditional Kalman filters (left of Figure 2) require explicit parametric models of the system's complex dynamics; however, it is difficult to specify the prior and likelihood probability densities explicitly to complete the Bayesian inference step. Therefore, Bayesian state estimation directly in the observation space is cumbersome. To bypass these parametric constraints, the Kernel Kalman Rule (KKR) (Gebhardt et al., 2019) is adopted. This method leverages the covariance operators established in Section 2.1 to perform

---

[1]A detailed matrix-based derivation of ELTO is given in Algorithm 3, Appendix A. We also compare the performance against model-based AEKF baselines in Table 1.

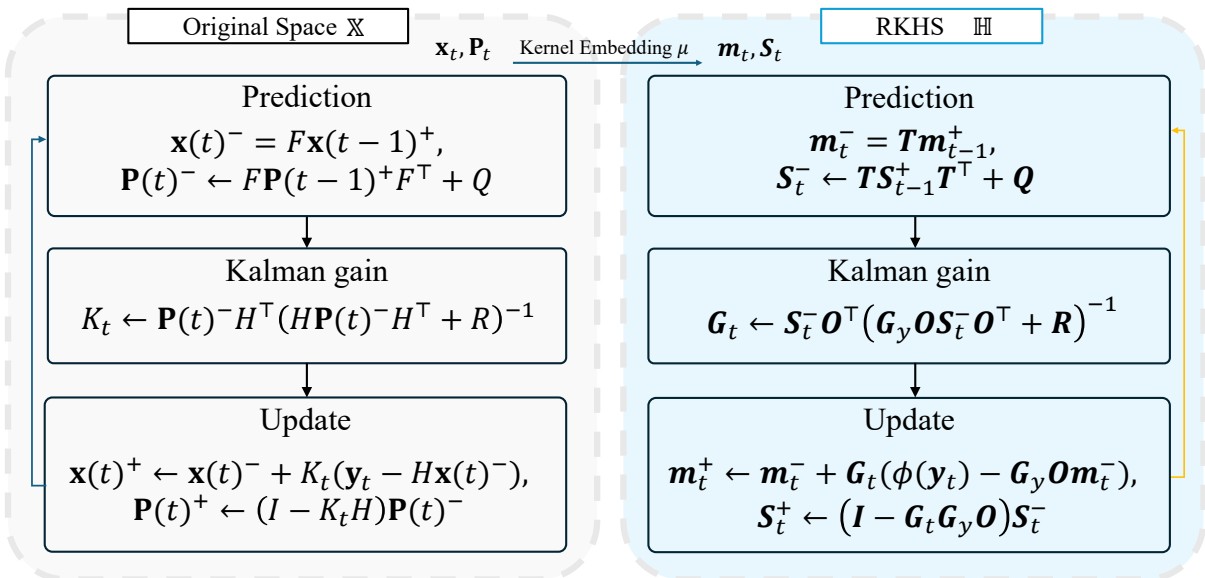

Figure 2: Comparison of Kalman filtering in the original space and in an RKHS. Through kernel embedding $\mu$, the state vector $\mathbf{x}(t)$ and its covariance $\mathbf{P}(t)$ are lifted to the RKHS mean $\boldsymbol{m}_t$ and covariance operator $\boldsymbol{S}_t$, with $\bullet^-$ and $\bullet^+$ denoting the prior and posterior. We utilize linear matrices $F$ and $H$ (instead of nonlinear transitions $p_{tr}$ and $p_{ob}$) to provide better intuition.

Bayesian updates directly in the RKHS (right of Figure 2), thereby avoiding the need for explicit density specifications (Fukumizu et al., 2013).

**Kernel Kalman Rule.** Given a training dataset $\{(\tilde{\boldsymbol{x}}_i, \boldsymbol{x}_i, \boldsymbol{y}_i)\}_{i=1}^N$, (Gebhardt et al., 2019) formulated the KKR to compute the empirical transfer and observable operators. $\tilde{\boldsymbol{x}}_i$ represents the preceding state (corresponding to $\mathbf{x}(t)$ in SSM) and $\boldsymbol{x}_i$ denotes the current state (corresponding to $\mathbf{x}(t+1)$ in the SSM) with its associated measurement $\boldsymbol{y}_i$ [2]. Let $k_x(\cdot, \cdot)$ and $k_y(\cdot, \cdot)$ be the kernel functions for the state and observation spaces, associated with feature maps $\psi(\cdot)$ and $\phi(\cdot)$, respectively. The feature matrices are defined as:

$$\boldsymbol{\Psi} = [\psi(\boldsymbol{x}_1), \ldots, \psi(\boldsymbol{x}_N)], \quad \boldsymbol{\Phi} = [\phi(\boldsymbol{y}_1), \ldots, \phi(\boldsymbol{y}_N)]. \tag{2}$$

Subsequently, the kernel matrices (Gram matrices) can be computed as:

$$\boldsymbol{G}_x = \boldsymbol{\Psi}^\top \boldsymbol{\Psi}, \quad \boldsymbol{G}_{yx} = \boldsymbol{\Phi}^\top \boldsymbol{\Psi}, \quad \boldsymbol{G}_{\tilde{x}} = \boldsymbol{\Psi}_1^\top \boldsymbol{\Psi}_1, \quad \boldsymbol{G}_{\tilde{x}x} = \boldsymbol{\Psi}_1^\top \boldsymbol{\Psi}_2, \quad \boldsymbol{G}_y = \boldsymbol{\Phi}^\top \boldsymbol{\Phi}, \tag{3}$$

where $\boldsymbol{\Psi}_1 := \boldsymbol{\Psi}_{:,1:N-1}$ and $\boldsymbol{\Psi}_2 := \boldsymbol{\Psi}_{:,2:N}$. Based on these kernel matrices, the transition matrix $\boldsymbol{T}$ and observation matrix $\boldsymbol{O}$ are derived as:

$$\boldsymbol{T} = (\boldsymbol{G}_{\tilde{x}} + \epsilon_t \boldsymbol{I})^{-1} \boldsymbol{G}_{\tilde{x}x}, \quad \boldsymbol{O} = (\boldsymbol{G}_x + \epsilon_o \boldsymbol{I})^{-1} \boldsymbol{G}_{yx}^\top, \tag{4}$$

where $\epsilon_t, \epsilon_o > 0$. $\boldsymbol{T}$ and $\boldsymbol{O}$ serve as the empirical representations of the conditional embedding operators $\mathcal{C}_{\mathbf{x}|\tilde{\mathbf{x}}}$ and $\mathcal{C}_{\mathbf{y}|\mathbf{x}}$, which govern the evolution of the transition and observation densities in the RKHS.

**Empirical Kernel Kalman Filters** Sequential estimation is performed by recursively calculating the posterior state through iterative prediction and update steps. To implement these steps in the RKHS using the finite-dimensional operator representations derived above, the mean map $\hat{\mu}_{\mathbf{x}(t)}$ and the covariance

---

[2]The data-driven construction of the state pairs $(\tilde{\boldsymbol{x}}_i, \boldsymbol{x}_i)$ is performed using the spectral learning algorithm described in Appendix A.

operator $\hat{\mathcal{C}}_{\mathbf{x}(t)}$ are tracked through a weight vector $\boldsymbol{m}_t \in \mathbb{R}^N$ and a weight matrix $\boldsymbol{S}_t \in \mathbb{R}^{N \times N}$ as follows:

$$\hat{\mu}_{\mathbf{x}(t)}^- = \boldsymbol{\Psi} \boldsymbol{m}_t^- \quad \text{and} \quad \hat{\mathcal{C}}_{\mathbf{x}(t)}^- = \boldsymbol{\Psi} \boldsymbol{S}_t^- \boldsymbol{\Psi}^\top, \tag{5}$$

where the a priori and a posteriori belief states are denoted as $\bullet^-$ and $\bullet^+$, respectively. The operators are updated in the RKHS map based on linear operations on these weights. The prediction step corresponds to:

$$\hat{\mu}_{\mathbf{x}(t)}^- = \hat{\mathcal{C}}_{\mathbf{x}|\tilde{\mathbf{x}}} \hat{\mu}_{\mathbf{x}(t-1)}^+ = \boldsymbol{\Psi} \boldsymbol{T} \boldsymbol{m}_{t-1}^+ \quad \Leftrightarrow \quad \boldsymbol{m}_t^- = \boldsymbol{T} \boldsymbol{m}_{t-1}^+,$$
$$\hat{\mathcal{C}}_{\mathbf{x}(t)}^- = \hat{\mathcal{C}}_{\mathbf{x}|\tilde{\mathbf{x}}} \hat{\mathcal{C}}_{\mathbf{x}(t-1)}^+ \hat{\mathcal{C}}_{\mathbf{x}|\tilde{\mathbf{x}}}^\top + \mathcal{C}_V \quad \Leftrightarrow \quad \boldsymbol{S}_t^- = \boldsymbol{T} \boldsymbol{S}_{t-1}^+ \boldsymbol{T}^\top + \boldsymbol{Q}. \tag{6}$$

Here, $\mathcal{C}_V$ and $\boldsymbol{Q}$ represent the process noise covariance in operator and matrix form, respectively. The update step, involving the Kalman gain, is similarly formulated as:

$$\hat{\mu}_{\mathbf{x}(t)}^+ = \hat{\mu}_{\mathbf{x}(t)}^- + \mathcal{G}_t(\phi(\boldsymbol{y}_t) - \mathcal{C}_{\mathbf{y}|\mathbf{x}} \hat{\mu}_{\mathbf{x}(t)}^-),$$
$$\hat{\mathcal{C}}_{\mathbf{x}(t)}^+ = \hat{\mathcal{C}}_{\mathbf{x}(t)}^- - \mathcal{G}_t \mathcal{C}_{\mathbf{y}|\mathbf{x}} \hat{\mathcal{C}}_{\mathbf{x}(t)}^-,$$
$$\mathcal{G}_t = \hat{\mathcal{C}}_{\mathbf{x}(t)}^- \mathcal{C}_{\mathbf{y}|\mathbf{x}}^\top (\mathcal{C}_{\mathbf{y}|\mathbf{x}} \hat{\mathcal{C}}_{\mathbf{x}(t)}^- \mathcal{C}_{\mathbf{y}|\mathbf{x}}^\top + \mathcal{C}_W)^{-1}. \tag{7}$$

$\mathcal{G}_t$ is the Kalman gain operator. $\mathcal{C}_W$ denotes the measurement noise covariance in operator form, whereas $\boldsymbol{R}$ (introduced later) denotes its matrix form. We provide the explicit finite-dimensional update equations in Section 3.2 (Equation 11-13), where they are integrated with our proposed structured noise parameterization.

## 3 Structured Noise Model Adaptation for Sequential Bayesian Filtering

The key limitation of existing Kalman filters is their reliance on dense, unstructured covariance models, which make the direct optimization ill-posed for non-stationary processes. In this section, we incorporate structured noise model adaptation to the data-driven optimization of the KKF, proposing the ELTO-AKF. Within our proposed method, KKF formulates the sequential filtering process given the data-driven system operators obtained from the spectral learning. Constructed by a new sparse structure parameterization detailed in Section 3.1, the structured noise model couples the data-driven optimization for tracking global noise statistics in noisy environments (Section 3.2) with dynamic parameter adaptation for enhancing robustness in non-stationary environments (Section 3.3).

### 3.1 Sparse Structure Parameterization

We introduce the scalar-block (SB) structure to facilitate the tractable learning of noise covariance matrices.

**Definition 1.** *Let a matrix $\boldsymbol{A} \in \mathbb{R}^{N \times N}$ be partitioned into $k \times k$ sub-blocks. We define a **scalar-block matrix** $\boldsymbol{A}_\theta^S$ as a matrix where each sub-block is a scaled identity matrix parameterized by a scalar $\theta_{ij} \in \mathbb{R}$:*

$$\boldsymbol{A}_\theta^S = \begin{pmatrix} \theta_{11} \cdot \boldsymbol{I} & \theta_{12} \cdot \boldsymbol{I} & \dots & \theta_{1k} \cdot \boldsymbol{I} \\ \theta_{21} \cdot \boldsymbol{I} & \theta_{22} \cdot \boldsymbol{I} & \dots & \theta_{2k} \cdot \boldsymbol{I} \\ \vdots & \vdots & \ddots & \vdots \\ \theta_{k1} \cdot \boldsymbol{I} & \theta_{k2} \cdot \boldsymbol{I} & \dots & \theta_{kk} \cdot \boldsymbol{I} \end{pmatrix}. \tag{8}$$

A significant challenge is to enforce the SB structure while guaranteeing the symmetric positive-definite (SPD) property required of a covariance matrix. To address this challenge, we adopt the Cholesky parameterization to transform a constrained covariance estimation into an unconstrained problem (Pinheiro & Bates, 1996). The technique represents the SB matrix via the factorization $\boldsymbol{A}_\theta^S = \boldsymbol{L} \boldsymbol{L}^\top$, where $\boldsymbol{L}$ is a block lower-triangular matrix whose sub-blocks are scaled identity matrices. Equivalently, under this Cholesky factorization, each covariance block satisfies $\theta_{ij} = \sum_{r=1}^{\min(i,j)} l_{ir} l_{jr}$. The SPD property of this SB structure is proved in the following.

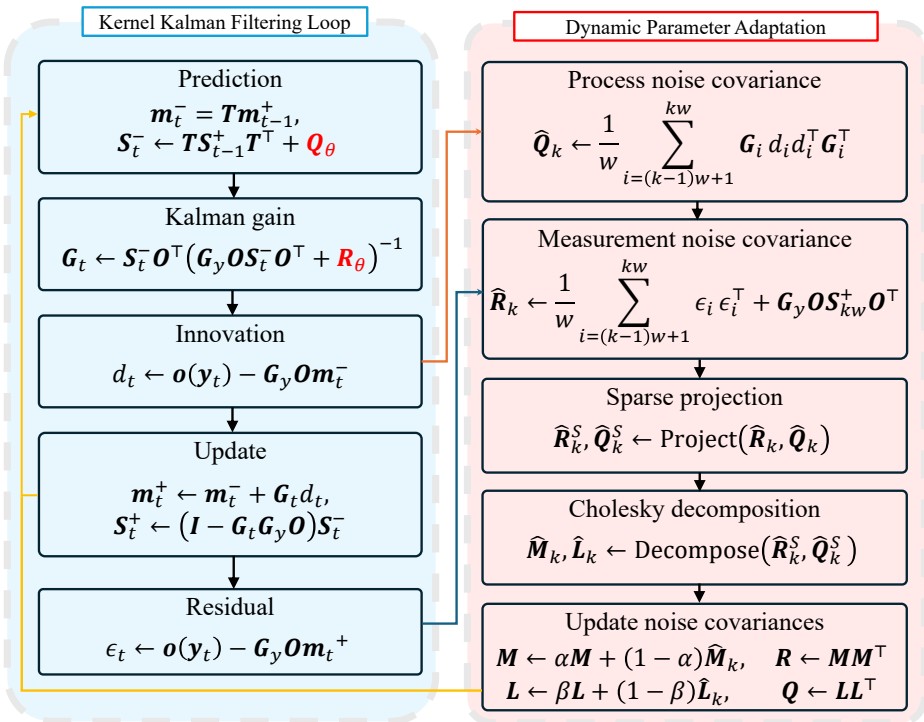

Figure 3: Schematic of the ELTO-AKF approach with dynamic parameter adaptation.

**Proposition 1.** *Let the factor matrix $\boldsymbol{L} \in \mathbb{R}^{N \times N}$ be a block lower-triangular matrix where each sub-block $\boldsymbol{L}_{ij} = l_{ij} \cdot \boldsymbol{I}$. If all diagonal scalars $l_{ii}$ are non-zero, then the matrix $\boldsymbol{A}$ constructed as $\boldsymbol{A} = \boldsymbol{L}\boldsymbol{L}^{\top}$ is a symmetric positive-definite (SPD), scalar-block matrix.*

**Proof.** Symmetry and the scalar-block structure follow directly from the construction $\boldsymbol{A} = \boldsymbol{L}\boldsymbol{L}^{\top}$ and the properties of block matrix multiplication. For positive definiteness, we consider the quadratic form $x^{\top}\boldsymbol{A}x = \|\boldsymbol{L}^{\top}x\|_2^2$. The condition $l_{ii} \neq 0$ for all $i$ ensures that the block lower-triangular matrix $\boldsymbol{L}$ is invertible. Since $\boldsymbol{L}$ is invertible, $\boldsymbol{L}^{\top}x \neq 0$ for any non-zero vector $\boldsymbol{x}$, which guarantees that the norm $\|\boldsymbol{L}^{\top}x\|_2^2$ is positive. Thus, $\boldsymbol{A}$ is positive-definite. $\square$

The SB parameterization, defined by the set of learnable scalars $\{\theta_{ij}\}$, provides an expressive yet computationally tractable representation of the noise covariance matrices, serving as a crucial component for establishing the structured noise adaptation of our ELTO-AKF.

## 3.2 Data-driven Optimization of Kernel Kalman Filters

We detail the matrix-based implementation of our ELTO-AKF and formulate the optimization objective for learning an optimal, time-invariant noise model. The relationship between the abstract operators in Equation 6-7 and their concrete matrices is grounded in the KKF, where operators and state embeddings are represented as linear combinations of the feature mappings.

We implement the filtering process using the kernel Kalman rule (Gebhardt et al., 2019), employing the system matrices $\boldsymbol{T}$ and $\boldsymbol{O}$ derived empirically from Gram matrices via the spectral learning algorithm detailed in Appendix A. ELTO-based filters (Ke et al., 2025) originally restrict noise covariances to static, identity-based matrices (i.e., $\boldsymbol{Q} = \epsilon_q \boldsymbol{I}$ and $\boldsymbol{R} = \epsilon_r \boldsymbol{I}$). This non-adaptive design is insufficient for capturing time-varying dynamics, leading to suboptimal Kalman gains $\boldsymbol{G}_t$ and degraded filtering performance in non-stationary environments. To address this problem, we therefore parameterize the noise covariance matrices as learnable, structured matrices (i.e., $\boldsymbol{Q}_\theta$ and $\boldsymbol{R}_\theta$) using the SB structure.

The sequential filtering process proceeds for $t = 1, \ldots, T$ as follows:

Prediction:

$$\boldsymbol{m}_t^- = \boldsymbol{T}\boldsymbol{m}_{t-1}^+, \tag{9}$$

$$\boldsymbol{S}_t^- = \boldsymbol{T}\boldsymbol{S}_{t-1}^+\boldsymbol{T}^\top + \boldsymbol{Q}_\theta. \tag{10}$$

Update:

$$\boldsymbol{G}_t = \boldsymbol{S}_t^-\boldsymbol{O}^\top(\boldsymbol{G}_y\boldsymbol{O}\boldsymbol{S}_t^-\boldsymbol{O}^\top + \boldsymbol{R}_\theta)^{-1}, \tag{11}$$

$$\boldsymbol{m}_t^+ = \boldsymbol{m}_t^- + \boldsymbol{G}_t(\boldsymbol{o}(\boldsymbol{y}_t) - \boldsymbol{G}_y\boldsymbol{O}\boldsymbol{m}_t^-), \tag{12}$$

$$\boldsymbol{S}_t^+ = (\boldsymbol{I} - \boldsymbol{G}_t\boldsymbol{G}_y\boldsymbol{O})\boldsymbol{S}_t^-. \tag{13}$$

Here, we define the feature map $\boldsymbol{o}(\boldsymbol{y}_t)$ from the subsequence $\boldsymbol{Y}_{1:N}$ of the training observation $\boldsymbol{Y} = [\boldsymbol{y}_1, \ldots, \boldsymbol{y}_T]$, where $N = T - 2h + 2$ and $h$ is the window size. This map is realized via the kernel trick as a vector of kernel evaluations: $\boldsymbol{o}(\boldsymbol{y}_t) := [k_y(\boldsymbol{y}_1, \boldsymbol{y}_t), \ldots, k_y(\boldsymbol{y}_N, \boldsymbol{y}_t)]^\top$. After obtaining the posterior state estimate $(\boldsymbol{m}_t^+, \boldsymbol{S}_t^+)$, the pre-image step maps the estimate back into the original observation space. The following pre-image estimate $\hat{\boldsymbol{\eta}}_t$ and $\hat{\boldsymbol{\Sigma}}_t$ can be compared with the measurement (Gebhardt et al., 2019).

$$\hat{\boldsymbol{\eta}}_t = \boldsymbol{Y}_{1:N}\boldsymbol{O}\boldsymbol{m}_t^+ \tag{14}$$

$$\hat{\boldsymbol{\Sigma}}_t = \boldsymbol{Y}_{1:N}\boldsymbol{O}\boldsymbol{S}_t^+\boldsymbol{O}^\top\boldsymbol{Y}^\top \tag{15}$$

The sequence of estimates $\{\hat{\boldsymbol{\eta}}_t\}_{t=1}^T$ depends on the noise parameters $\theta$ through the matrices $\boldsymbol{Q}_\theta$ and $\boldsymbol{R}_\theta$, which influence the Kalman gain $\boldsymbol{G}_t$ and covariance propagation. Since filter performance is impacted by these matrices and standard estimation relies on potentially violated assumptions (Wang, 1999), directly minimizing $\theta$ to minimize the end-to-end estimation error offers a data-driven approach to finding an optimal time-invariant model with the best overall robustness for a given distribution (Greenberg et al., 2023). Our objective is the mean squared error (MSE):

$$\theta^* = \underset{\theta}{\operatorname{argmin}}\frac{1}{T}\sum_{t=1}^T ||\hat{\boldsymbol{\eta}}_t - \boldsymbol{y}_t^{\text{val}}||^2. \tag{16}$$

Here, $\boldsymbol{Y}^{\text{val}} = [\boldsymbol{y}_1^{\text{val}}, \ldots, \boldsymbol{y}_T^{\text{val}}]$ is a split of the full observation data different from training data $\boldsymbol{Y}$ that is used for estimating the system operators from Algorithm 3. $\boldsymbol{Y}^{\text{val}}$ lets us evaluate the generalizability of $\boldsymbol{T}$ and $\boldsymbol{O}$ properly, and also find the optimal $\theta^*$ of the noise covariance matrices for the filter.

Following the monotonicity property of the RBF kernel proven in Appendix B, minimizing the squared error in observation space (Equation 16) shares the same minimizer as minimizing the filter's innovation residual $\boldsymbol{o}(\boldsymbol{y}_t) - \boldsymbol{G}_y\boldsymbol{O}\boldsymbol{m}_t^-$ in the RKHS (cf. Equation 12), providing a principled (rather than purely heuristic) optimization target. Unlike traditional parameter estimation in SSMs, which, for example, relies on maximum likelihood estimation via particle methods (Kantas et al., 2015), our RKHS formulation bypasses explicit probability density functions. To numerically solve this optimization objective, we therefore employ CMA-ES (Hansen, 2006), a derivative-free evolution strategy well suited for minimizing non-convex functions with a small number of parameters. Algorithm 1 finds the optimal noise model for the observation distribution, ensuring robust performance in noisy, non-stationary environments [3].

### 3.3 Dynamic Parameter Adaptation

We detail the dynamic parameter adaptation for non-stationary environments where noise levels or dynamic characteristics of the system change abruptly over time, as shown in Figure 4c. In such scenarios, fixed covariance matrices fail to capture sudden dynamic shifts. We detect these changes by tracking pre-fit and post-fit errors: the innovation and the residual. A spike in innovation indicates a shift in the system dynamics, and a large posterior residual suggests that the updated state still struggles to reconcile with (potentially

---

[3]The code is available online at `https://github.com/keeenda/ELTO-AKF`.

---

**Algorithm 1** ELTO-based Kalman Filtering with Structured Noise Model Adaptation

---

1: **Input:** training data $\boldsymbol{Y} = [\boldsymbol{y}_1, \ldots, \boldsymbol{y}_T]$, validation data $\boldsymbol{Y}^{\text{val}} = [\boldsymbol{y}_1^{\text{val}}, \ldots, \boldsymbol{y}_T^{\text{val}}]$, window size $w$, memory factors $(\alpha, \beta)$, and randomly initialized parameters $\epsilon_q, \epsilon_r$ or $\epsilon_l, \epsilon_m$.
2: *// — Initialize ELTO —*
3: Compute transfer and observable matrices $\boldsymbol{T}$ and $\boldsymbol{O}$ following Algorithm 3 using $\boldsymbol{Y}$,
4: Initialize mean $\boldsymbol{m}_0$ and variance $\boldsymbol{S}_0$
5: *// — Initialize noise covariance matrices following Equation 8 —*
6: **if** $\alpha = \beta = 1$ **then**
7:     Parameterize $\boldsymbol{Q}_0$ and $\boldsymbol{R}_0$ with SB structure using $\theta_{ij} = \epsilon_q$ or $\epsilon_r$, respectively.
8: **else**
9:     Parameterize $\boldsymbol{L}_0, \boldsymbol{M}_0$ with SB structure using $\theta_{ij} = \epsilon_l$ or $\epsilon_m$, respectively, for $i \geq j$
10:     $\boldsymbol{L} \leftarrow \boldsymbol{L}_0, \boldsymbol{M} \leftarrow \boldsymbol{M}_0$
11:     $\boldsymbol{Q} \leftarrow \boldsymbol{L}\boldsymbol{L}^\top, \ \boldsymbol{R} \leftarrow \boldsymbol{M}\boldsymbol{M}^\top$
12: **end if**
13: *// — Derivative-free optimization by CMA-ES —*
14: Compute the pre-image estimates $[\hat{\boldsymbol{\eta}}_1, \ldots, \hat{\boldsymbol{\eta}}_T], [\hat{\boldsymbol{\Sigma}}_1, \ldots, \hat{\boldsymbol{\Sigma}}_T]$ and the corresponding $\theta^*$ following Algorithm 2 using $\boldsymbol{Y}^{\text{val}}$
15: **Output:** $\theta^*, [\hat{\boldsymbol{\eta}}_1, \ldots, \hat{\boldsymbol{\eta}}_T]$ and $[\hat{\boldsymbol{\Sigma}}_1, \ldots, \hat{\boldsymbol{\Sigma}}_T]$

---

unreliable) incoming sensor data. Specifically, we employ the analytical update rules of the adaptive extended Kalman filter (AEKF) (Akhlaghi et al., 2017) to properly handle non-stationary processes. The innovation and residual are estimated as follows:

$$
\begin{array}{ccc}
\text{Operator form} & \Leftrightarrow & \text{Matrix form} \\
\phi(\boldsymbol{y}_t) - \mathcal{C}_{\mathbf{y}|\mathbf{x}}\hat{\mu}_{\mathbf{x}(t)}^- & \Leftrightarrow & d_t = \boldsymbol{o}(\boldsymbol{y}_t) - \boldsymbol{G}_y\boldsymbol{O}\boldsymbol{m}_t^-, \\
\phi(\boldsymbol{y}_t) - \mathcal{C}_{\mathbf{y}|\mathbf{x}}\hat{\mu}_{\mathbf{x}(t)}^+ & \Leftrightarrow & \epsilon_t = \boldsymbol{o}(\boldsymbol{y}_t) - \boldsymbol{G}_y\boldsymbol{O}\boldsymbol{m}_t^+.
\end{array}
\tag{17}
$$

The AEKF update uses memory factors $(\alpha, \beta)$ to dynamically adapt the estimates of $\boldsymbol{Q}$ and $\boldsymbol{R}$[4]:

$$
\begin{aligned}
\boldsymbol{R}_t &= \alpha\boldsymbol{R}_{t-1} + (1-\alpha)(\epsilon_t\epsilon_t^\top + \boldsymbol{G}_y\boldsymbol{O}\boldsymbol{S}_t^+\boldsymbol{O}^\top), \\
\boldsymbol{Q}_t &= \beta\boldsymbol{Q}_{t-1} + (1-\beta)(\boldsymbol{G}_t d_t d_t^\top \boldsymbol{G}_t^\top).
\end{aligned}
\tag{18}
$$

However, directly updating covariance matrices faces hurdles with positive-definiteness and high dimensionality. As shown in Figure 3, our ELTO-AKF approach circumvents these issues by operating on the Cholesky factors ($\boldsymbol{L}$ and $\boldsymbol{M}$), leveraging the advantages of the SB parameterization. This approach guarantees that $\boldsymbol{Q}$ and $\boldsymbol{R}$ are SPD matrices and maintains their tractability. Without the SPD guarantee for the noise covariance matrices, the performance of traditional Kalman filtering can degrade significantly (Greenberg et al., 2023).

The adaptive process begins with initial Cholesky factors $\boldsymbol{L}_0$ and $\boldsymbol{M}_0$, defining $\boldsymbol{Q}_0 = \boldsymbol{L}_0\boldsymbol{L}_0^\top$ and $\boldsymbol{R}_0 = \boldsymbol{M}_0\boldsymbol{M}_0^\top$. For the periodic updates, we employ a sparse projection operator $\text{SP}(\cdot)$, defined in two stages. First, we enforce the SB structure on the dense estimate $\boldsymbol{A}$ (representing either $\hat{\boldsymbol{Q}}_k$ or $\hat{\boldsymbol{R}}_k$) by averaging its block traces: $\theta_{ij} = \frac{\text{tr}(\boldsymbol{A}_{ij})}{\text{rank}(\boldsymbol{A}_{ij})}$, yielding an intermediate matrix $\tilde{\boldsymbol{A}}$. Second, to guarantee the SPD property required for Cholesky factorization, we perform the rectification $\boldsymbol{A}^S = \boldsymbol{V}\tilde{\boldsymbol{\Lambda}}\boldsymbol{V}^\top$, where $\tilde{\boldsymbol{A}} = \boldsymbol{V}\boldsymbol{\Lambda}\boldsymbol{V}^\top$ is the eigen-decomposition and $\tilde{\boldsymbol{\Lambda}} = \text{diag}\big(\max(\lambda_i, \xi)\big)_{i=1}^N$ with $\xi > 0$ denoting a minimum stability threshold. The resulting positive-definite matrix $\boldsymbol{A}^S$ is decomposed into Cholesky factors, which are then updated using the memory terms, as detailed in Algorithm 2.

---

[4] For example, a larger memory factor puts more weights on the previous state, helping to recover to overall trend of the system dynamics and not get easily damaged by the abrupt change, as shown in Figure 4c.

---

**Algorithm 2** Kernel Kalman Filtering with Dynamic Parameter Adaptation

---

1: **Input:** $\boldsymbol{T}$, $\boldsymbol{O}$, $\boldsymbol{m}_0$, $\boldsymbol{M}$, $\boldsymbol{L}$ , $\boldsymbol{S}_0$, $\boldsymbol{Q}_\theta$, $\boldsymbol{R}_\theta$ from Algorithm 1, $\boldsymbol{Y}^{\text{val}} = [\boldsymbol{y}_1^{\text{val}}, \ldots, \boldsymbol{y}_T^{\text{val}}]$, window size $w$, and memory factors $(\alpha, \beta)$.

2: **for** $t = 1, \ldots, T$ **do**

3:    *//— Prediction —*

4:    $\boldsymbol{m}_t^- \leftarrow \boldsymbol{T}\boldsymbol{m}_{t-1}^+, \quad \boldsymbol{S}_t^- \leftarrow \boldsymbol{T}\boldsymbol{S}_{t-1}^+\boldsymbol{T}^\top + \boldsymbol{Q}_\theta$

5:    *//— Update —*

6:    $\boldsymbol{G}_t \leftarrow \boldsymbol{S}_t^-\boldsymbol{O}^\top(\boldsymbol{G}_y\boldsymbol{O}\boldsymbol{S}_t^-\boldsymbol{O}^\top + \boldsymbol{R}_\theta)^{-1}$                    ▷ *Kalman gain*

7:    $d_t \leftarrow \boldsymbol{o}(\boldsymbol{y}_t) - \boldsymbol{G}_y\boldsymbol{O}\boldsymbol{m}_t^-$                    ▷ *Innovation*

8:    $\boldsymbol{m}_t^+ \leftarrow \boldsymbol{m}_t^- + \boldsymbol{G}_t d_t, \quad \boldsymbol{S}_t^+ \leftarrow (\boldsymbol{I} - \boldsymbol{G}_t\boldsymbol{G}_y\boldsymbol{O})\boldsymbol{S}_t^-$

9:    $\epsilon_t \leftarrow \boldsymbol{o}(\boldsymbol{y}_t) - \boldsymbol{G}_y\boldsymbol{O}\boldsymbol{m}_t^+$                    ▷ *Residual*

10:    *//— Dynamic Parameter Adaptation —*

11:    $k \leftarrow \lfloor t/w \rfloor, \quad r \leftarrow t \bmod w$

12:    **if** $r = 0$ and not $\alpha = \beta = 1$ **then**        ▷ *In this condition, $\boldsymbol{Q}$, $\boldsymbol{R}$ are time-variant. $\theta$ is abbreviated.*

13:       $\hat{\boldsymbol{R}}_k \leftarrow \frac{1}{w}\sum_{i=(k-1)w+1}^{kw} \epsilon_i\epsilon_i^\top + \boldsymbol{G}_y\boldsymbol{O}\boldsymbol{S}_{kw}^+\boldsymbol{O}^\top, \quad \hat{\boldsymbol{Q}}_k \leftarrow \frac{1}{w}\sum_{i=(k-1)w+1}^{kw} \boldsymbol{G}_i d_i d_i^\top \boldsymbol{G}_i^\top$

14:       $\hat{\boldsymbol{R}}_k^S \leftarrow \text{SP}(\hat{\boldsymbol{R}}_k), \quad \hat{\boldsymbol{Q}}_k^S \leftarrow \text{SP}(\hat{\boldsymbol{Q}}_k)$                    ▷ *Sparse projection into SB structure*

15:       $\hat{\boldsymbol{M}}_k \leftarrow \text{Decompose}(\hat{\boldsymbol{R}}_k^S), \quad \hat{\boldsymbol{L}}_k \leftarrow \text{Decompose}(\hat{\boldsymbol{Q}}_k^S)$                    ▷ *Cholesky decomposition*

16:       $\boldsymbol{M} \leftarrow \alpha\boldsymbol{M} + (1-\alpha)\hat{\boldsymbol{M}}_k, \quad \boldsymbol{L} \leftarrow \beta\boldsymbol{L} + (1-\beta)\hat{\boldsymbol{L}}_k$

17:       $\boldsymbol{R} \leftarrow \boldsymbol{M}\boldsymbol{M}^\top, \quad \boldsymbol{Q} \leftarrow \boldsymbol{L}\boldsymbol{L}^\top$

18:    **end if**

19:    *//— Projection back to the original space —*

20:    $\hat{\boldsymbol{\eta}}_t, \hat{\boldsymbol{\Sigma}}_t \leftarrow \text{pre\_image}(\boldsymbol{m}_t^+, \boldsymbol{S}_t^+)$                    ▷ *$\hat{\boldsymbol{\eta}}_t$ is a short-hand notation for $\hat{\boldsymbol{\eta}}_t(\theta)$.*

21: **end for**

22: $\theta^* = \underset{\theta}{\text{argmin}} \frac{1}{T}\sum_{t=1}^{T}\|\hat{\boldsymbol{\eta}}_t - \boldsymbol{y}_t^{\text{val}}\|^2$                    ▷ *During the test phase, we perform inference using $\theta^*$.*

23: **Output:** $\theta^*$, $[\hat{\boldsymbol{\eta}}_1, \ldots, \hat{\boldsymbol{\eta}}_T]$ and $[\hat{\boldsymbol{\Sigma}}_1, \ldots, \hat{\boldsymbol{\Sigma}}_T]$

---

## 4 Numerical Experiments

### 4.1 Denoising the Nonlinear Pendulum System

To evaluate the performance of our proposed method in a standard filtering context, we first examined its denoising ability on a simulated pendulum task. The experimental setup follows the one described in (Gebhardt et al., 2019), based on a simulated single pendulum system[5]. For each simulation run, the initial angle $q_0$ and angular velocity $\dot{q}_0$ were uniformly sampled from the ranges $[-0.25\pi, 0.25\pi]$ and $[-2\pi, 2\pi]$ rad/s, respectively. The dynamics were simulated at 10,000 Hz subject to a normally distributed process noise with standard deviation $n_p$. Observations of the joint positions were collected at 10 Hz, and corrupted by an additive Gaussian noise with standard deviation $n_o$ centered on the true angle $q_t$. For our evaluations, we defined 3 specific noise regimes: Default setting ($n_p = 0.1, n_o = 0.01$), high process noise ($n_p = 0.2, n_o = 0.1$) and high observation noise ($n_p = 0.1, n_o = 0.1$).

The generated data were partitioned into a training sequence of 1500 and a test sequence of 300 time steps. The spectral learning architecture was trained using the Adam optimizer with a learning rate of $10^{-3}$ and configured with a window size of 5. For our proposed structured noise model, which used a diagonal parameterization with $k_q = k_r = 5$ blocks, the optimal noise variances were tuned using CMA-ES.

We benchmark ELTO-AKF against four fundamentally different baselines. Constant Velocity AEKF (CV) and linear AEKF (Linear) rely on analytical priors or assumed linear system dynamics. Sampling-based and neural-aided Kalman filters (Hu et al., 2025; Loo et al., 2024), such as the Sigma-Point Kalman Filter (SPKF) and CKFNet, avoid explicit Jacobians via spatial sampling and often integrate recurrent networks to adapt to unmodeled dynamics. The Kernel Kalman Rule (KKR) (Gebhardt et al., 2019) and its efficient subsampled approximation (SubKKR) perform non-parametric updates on RKHSs while leaving the latent

---

[5]The simulation was performed using the code available at `https://github.com/gregorgebhardt/pyKKR`.

state $\boldsymbol{x}$ model-based. Finally, operator-based ELTO-KF learns from observations but uses fixed identity-based covariance matrices ($\boldsymbol{Q} = \epsilon_q \boldsymbol{I}$, $\boldsymbol{R} = \epsilon_r \boldsymbol{I}$).

Table 1a reports the mean squared error (MSE) averaged over 5 trials. Under default noise, the model-based AEKF (CV) achieves the lowest error since its analytical priors align well with simple dynamics. However, as noise increases, AEKF and SPKF errors rise considerably due to fixed priors and sparse sampling. While neural-aided architectures like CKFNet show promise in standard settings, they often encounter numerical divergence under extreme noise conditions, as unconstrained neural predictions struggle to preserve the symmetric positive-definite (SPD) property of covariance matrices. Similarly, supervised KKR/SubKKR methods struggle with complex noise distributions. In contrast, the data-driven ELTO-AKF is competitive in the default setting and outperforms baselines under high process and high observation noise. The Scalar-Block parameterization and its low-dimensional Cholesky updates effectively overcome static noise limitations and structurally guarantee stability in dynamic environments. A conceptual comparison of these filtering methods across key properties is summarized in Table 1b, showing that our data-driven ELTO-AKF is the only method designed to be adaptive for non-stationary processes while guaranteeing the SPD structure.

Table 1: Comparison of sequential state estimation methods.

(a) Denoising performance (MSE $\times 10^{-3}$) on the pendulum dataset under various noise conditions.

| Noise Setting | Model-based AEKF | | Kernel- or Sampling-based | | | Operator-based | |
|---|---|---|---|---|---|---|---|
| | CV | Linear | SubKKR | KKR | SPKF | ELTO-KF | ELTO-AKF |
| Default | **0.0835** | 0.1321 | 2.7574 | 1.1185 | 1.6099 | 0.1010 | 0.1012 |
| High process noise | 0.1167 | 0.3079 | 3.9321 | 3.4061 | 3.473 | 0.2325 | **0.1123** |
| High observation noise | 9.181 | 10.309 | 13.192 | 15.797 | 12.218 | 11.278 | **8.881** |

(b) Conceptual comparison of filtering methodologies.

| Method | Data-driven operators | RKHS inference | Bypasses explicit PDF | Adaptable to non-stationarity | Guaranteed SPD structure |
|---|---|---|---|---|---|
| AEKF (CV/Linear) | × | × | × | ✓ | × |
| CKFNet | × | × | × | ✓ | × |
| SPKF | × | × | × | × | × |
| KKR/SubKKR | × | ✓ | ✓ | × | × |
| ELTO-KF | ✓ | ✓ | ✓ | × | × |
| **ELTO-AKF (Ours)** | ✓ | ✓ | ✓ | ✓ | ✓ |

## 4.2 Tracking Non-Stationary LiDAR Trajectories

We conducted a filtering experiment on a single, long synthetic trajectory to evaluate our method's ability to handle changing dynamics. Inspired by tracking problems in (Greenberg et al., 2023), our experimental setup is modified to test the model's ability to handle changing dynamics within a continuous data stream of LiDAR measurements[6]. Each trajectory consists of multiple segments of variable length, with each segment defined by constant radial and tangential accelerations sampled from a normal distribution. This process generates a path with alternating periods of straight-line motion and coordinated turns, creating the piecewise non-stationary dynamics. The base simulation uses initial positions in [-50, 50] per axis, an initial velocity magnitude in [1, 5], and radial and tangential acceleration standard deviations of 0.1 and 0.5. The generated trajectory provides the true path and corresponding noisy observations, which are then standardized. For a comprehensive evaluation, we created distinct datasets by systematically varying two key factors: noise levels (categorized as default, high, and very high) and trajectory segment lengths (categorized as default, small, and large)[7].

---

[6]The simulation was performed using code adapted from `https://github.com/ido90/Optimized-Kalman-Filter`.

[7]The specific parameter settings for each group are as follows: Noise Levels (in terms of `noise_r`, `noise_t`): Default (1, 0.5), High (2, 1.5), and Very High (5, 2.5); Segment Lengths (in terms of `n_intervals`, `int_len`): Default ((25, 30), (20, 25)), Small ((50, 60), (10, 12)), and Large ((10, 15), (40, 50)).

The first 50% of the trajectory is used for training, and the other half is used for testing. To evaluate the model under different practical constraints, we designed two testing regimes in Table 2. In the Full Optimization setting, we search for the optimal $\epsilon_q$ and $\epsilon_r$ for the initial noise covariance matrices ($\boldsymbol{Q}_0$ and $\boldsymbol{R}_0$) using CMA-ES, while keeping $\alpha = \beta = 1$ fixed throughout the filtering process. In the Pure Adaptation setting, we ablate the optimization phase and rely purely on dynamic parameter adaptation (refer to Algorithm 2), starting from naive initializations with memory factors $\alpha = \beta = 0.9$. Note that kernel parameters (e.g., lengthscales) remain fixed across both settings. The filtering performance for these settings, along with evaluations under distribution shifts, is presented in Tables 2 and 3.

Table 2: Dynamic state estimation performance (MSE ($\times 10^{-2}$)) on piecewise non-stationary LiDAR trajectories. We compare the baseline ELTO-KF with our ELTO-AKF under two settings: full optimization and pure adaptation. In the former setting, the initial noise covariance matrices are tuned (using CMA-ES) with $\alpha = \beta = 1$. In the latter, they are tuned purely through dynamic parameter adaptation ($\alpha = \beta = 0.9$), without using CMA-ES.

| Parameter Group | Settings | Full Optimization | | Pure Adaptation | |
|---|---|---|---|---|---|
| | | ELTO-KF | ELTO-AKF ($\alpha = \beta = 1$) | ELTO-KF | ELTO-AKF ($\alpha = \beta = 0.9$) |
| Noise Levels | Default | 8.2825 | **1.9466** | 20.7132 | **8.8112** |
| | High | 12.5629 | **5.0497** | 33.9798 | **11.2288** |
| | Very High | 24.5529 | **13.0486** | 81.5785 | **35.0613** |
| Segment Lengths | Small | 9.2305 | **3.9511** | 22.2544 | **6.5787** |
| | Large | 12.9478 | **7.9840** | 26.2269 | **11.9133** |

Table 3: Robustness evaluation (MSE ($\times 10^{-1}$)) against distribution shifts. Models are evaluated on unseen noise levels during testing that differ from their training data. ELTO-AKF consistently demonstrates superior generalization under these mismatched conditions compared to the ELTO-KF baseline.

| Noise Levels | | Models | |
|---|---|---|---|
| Train | Test | ELTO-KF | ELTO-AKF |
| Default | High | 3.5604 | **1.0789** |
| Default | Very High | 7.2018 | **3.9934** |
| High | Very High | 8.1969 | **6.5971** |
| High | Default | 4.9959 | **3.1255** |
| Very High | Default | 4.9159 | **2.2983** |
| Very High | High | 3.6932 | **1.3865** |

As shown in Table 2, in the Full Optimization setting where parameters are finely tuned, ELTO-AKF significantly outperforms the standard ELTO-KF across all noise levels and trajectory lengths, validating the training efficacy of our proposed structure. Furthermore, in the Pure Adaptation setting—which tests the model without initial hyperparameter optimization—ELTO-AKF still maintains robust performance through dynamic parameter adaptation. Table 3 demonstrates that when faced with unseen noise and distribution shifts, ELTO-AKF also achieves a lower error compared to the baseline. Further ablation studies (provided in Appendix C) confirm that the proposed structure is crucial for maintaining numerical stability.

## 4.3 Scaling to High-Dimensional Lorenz-96 Systems

To evaluate the scalability and computational stability of our proposed method under varying dimensions, we conducted experiments using the standard Lorenz-96 chaotic system (Lorenz, 1996) (forcing constant $F = 8.0$) across dimensions $D \in \{5, 10, 100, 500, 1000\}$. Trajectories of 1000 steps were generated via RK4 integration (Press, 2007) ($\Delta t = 0.01$) and injected with varying levels of Gaussian observation noise ($\sigma_{\text{obs}} \in \{0.01, 0.1, 0.5\}$). The training was configured with 200 epochs, a batch size of 50, and a window $w = 5$. For ELTO configuration, the baseline was optimized via 50 CMA-ES iterations. For our ELTO-AKF,

the number of scalar blocks for the structured noise parameterization was dynamically scaled based on the system dimension, yielding $k \in \{2, 5, 10\}$ for the respective dimensions.

As summarized in Table 4, ELTO-AKF performs competitively with the baseline at low noise and outperforms it at moderate-to-high noise across most dimensions. More importantly, in several high-dimensional and high-noise settings (e.g., $D = 500$, $\sigma_{\text{obs}} = 0.1$ and $D = 1000$, $\sigma_{\text{obs}} = 0.5$), the ELTO-KF baseline diverges due to a loss of the SPD property, whereas the structured parameterization of ELTO-AKF inherently guarantees SPD and remains numerically stable. This indicates that, while the unstructured baseline can sometimes recover when initialization happens to remain well-conditioned, only the structured parameterization provides reliable scalability.

Table 4: Performance comparison (MSE) on the Lorenz-96 system. $D$ denotes the system dimension, and $\sigma_{obs}$ is the standard deviation of the observation noise. "N/A" indicates that the ELTO-KF baseline failed to converge in that configuration due to a loss of the SPD property; ELTO-AKF remained stable in every configuration.

| $D$ | Noise ($\sigma_{obs}$) = 0.01 | | Noise ($\sigma_{obs}$) = 0.1 | | Noise ($\sigma_{obs}$) = 0.5 | |
|---|---|---|---|---|---|---|
| | ELTO-KF | ELTO-AKF | ELTO-KF | ELTO-AKF | ELTO-KF | ELTO-AKF |
| 5 | **0.0415** | 0.0451 | 0.2304 | **0.0304** | 0.2076 | **0.2069** |
| 10 | 0.2243 | **0.1672** | 0.2368 | **0.1257** | **0.2814** | 0.3021 |
| 100 | **0.5032** | 0.5186 | 0.9489 | **0.5265** | 0.9890 | **0.5782** |
| 500 | 0.8760 | **0.8446** | N/A | **1.2722** | **0.9159** | 0.9252 |
| 1000 | 1.7682 | **1.6721** | 1.8227 | **1.6836** | N/A | **1.7384** |

## 4.4 Downstream Application: Data-Driven PDE Discovery

Our ELTO-AKF approach is well-suited for denoising noisy state variables using learned noise covariance matrices, improving the quality of the observational data. The improved data quality can, in turn, support downstream tasks of data-driven partial differential equation (PDE) discovery (Rudy et al., 2017), yielding more accurate recovery of governing PDEs. We aim to identify sparse governing PDEs parameterized in the following form: $u_t = \mathcal{N}(u, u_x, u_{xx}, \dots)$; where $\mathcal{N}$ denotes a hidden nonlinear operator involving the state variable $u$ and its spatial derivatives (e.g., $u_x, u_{xx}, \dots$). We assume $\mathcal{N}$ is expressed as a linear combination of a few active terms in the (sparse) governing equation. To identify the underlying PDE, we solve a sparse or best-subset regression problem constructed over a library of (independent) candidate terms, with the temporal derivative $u_t$ serving as the vector response for selecting the most relevant terms. Note that we only have access to noisy values of $u(x, t)$ on a discretized spatiotemporal domain. By applying numerical differentiation or weak-form computation (Reinbold et al., 2020), we can construct an overcomplete candidate library, setting up a system of equations for the regression problem. The best model (optimally balancing between approximation error and model complexity) that is most likely to represent the actual governing PDE is selected using information criteria (Thanasutives et al., 2024; 2025).

We aim to demonstrate that refining the observed states using our ELTO-AKF improves the accuracy of identified PDE coefficients. The parametric Burgers' equation is used as an example. The equation reads $u_t = \begin{bmatrix} u u_x & u_{xx} \end{bmatrix} \boldsymbol{\xi}$; where $\boldsymbol{\xi}$ is the true PDE coefficient vector, containing the kinematic viscosity (or diffusion coefficient) $\vartheta$. We study $\boldsymbol{\xi} = \begin{bmatrix} -1 & \vartheta \end{bmatrix}^T$; where $\vartheta = 0.1$ or $\frac{0.01}{\pi}$. In the latter case, the viscosity is so small that shock waves develop in the PDE solution, and hence, abrupt changes occur in the system dynamics. To generate $u(x, t)$, the clean state variable is perturbed with Gaussian noise drawn from $\frac{\epsilon \sigma}{100} \mathcal{N}(0, 1)$, where $\sigma$ denotes the standard deviation of the clean state variable. The noise level is controlled by the parameter $\epsilon$.

Our noise covariances are trained to denoise by mapping doubly noisy data to (noisy) observed data. Here, the doubly noisy $\tilde{u}(x, t) = u(x, t) + \mathcal{N}(0, \tilde{\sigma}^2)$ refers to noisy observations further perturbed with synthetic Gaussian noise, whose standard deviation $\tilde{\sigma}$ is estimated using a robust wavelet-based estimator (Donoho & Johnstone, 1994). A key practical advantage of using $\tilde{u}$ is that it does not need the clean state variable for training and validation. During the test phase, after obtaining the learned noise covariance matrices, we use them to denoise the observations. The resulting noise-reduced data are passed through the PDE

discovery method to estimate a sparse vector $\hat{\boldsymbol{\xi}}$ of the PDE coefficients, for which we calculate the average of its absolute percentage errors, each defined by $\mathcal{E}_j(\boldsymbol{\xi}, \hat{\boldsymbol{\xi}}) = \left| \frac{\xi_j - \hat{\xi}_j}{\xi_j} \right|$. Another evaluation metric we consider, which is less susceptible to errors in estimating small coefficients, is the L1 relative error: $\frac{||\boldsymbol{\xi} - \hat{\boldsymbol{\xi}}||_1}{||\boldsymbol{\xi}||_1}$.

Table 5: Comparison of PDE discovery results with and without denoising the noisy state variables. The mean squared error (MSE) implies the L2-distance between the data used for PDE discovery and the clean state-variable data. Better scores are shown in **bold**. The ELTO-AKF algorithm is run with $\alpha = \beta = 1$, and the correct viscosity is 0.1.

| Denoising | MSE ($\times 10^{-3}$) | Identified PDE | MAPE | L1 Relative Error ($\times 10^{-2}$) |
|---|---|---|---|---|
| N/A | 8.08 | $-0.959286uu_x + 0.076843u_{xx}$ | 13.61 | 5.81 |
| ELTO-AKF | **2.48** | $-0.940588uu_x + 0.098496u_{xx}$ | **3.72** | **5.54** |

Table 6: Comparison of denoising methods based on the ELTO-KF and our proposed ELTO-AKF with varying memory factors. Here, the correct viscosity is $\frac{0.01}{\pi}$. Note that the identified PDE is incorrect when no denoising is applied; therefore, its evaluation metrics are unavailable (N/A).

| Denoising | MSE ($\times 10^{-2}$) | Identified PDE | MAPE | L1 Relative Error ($\times 10^{-1}$) |
|---|---|---|---|---|
| N/A | N/A | $-0.464365uu_x + 0.000163u^2u_{xx}$ | N/A | N/A |
| ELTO-KF | 2.70 | $-0.677635uu_x + 0.006934u_{xx}$ | 75.03 | 3.25 |
| ELTO-AKF ($\alpha = \beta = 1$) | 2.76 | $-0.742664uu_x + 0.003891u_{xx}$ | **23.99** | 2.57 |
| ELTO-AKF ($\alpha = \beta = 0.7$) | **2.19** | $-0.787012uu_x + 0.004537u_{xx}$ | 31.92 | **2.14** |

To empirically demonstrate the effectiveness of our ELTO-AKF in denoising, we first conduct experiments on discovering the canonical Burgers' equation with viscosity $\vartheta = 0.1$ under a high noise level of $\epsilon = 50$. Table 5 shows that the denoised state-variable data resemble the clean data (as illustrated in Figure 4) more closely than the noisy observed data, as evidenced by the decreased MSE. Therefore, a higher-quality PDE—closer to the true governing form, as indicated by the low MAPE (around 4%) and the small L1 relative error—is identified. Then, we experiment on the viscous Burgers' equation exhibiting shock waves, where the PDE solution becomes discontinuous in space after a certain time, to reveal the full ELTO-AKF's capability in adapting to local dynamics by adjusting the memory factors. As shown in Table 6, the identified governing equation is incorrect if no denoising is applied and remains of low quality when denoising is performed using the ELTO-KF approach. The ELTO-AKF with adaptation to local dynamics achieves the best performance in terms of the MSE and the L1 relative error, whereas the ELTO-AKF with full memory attains the lowest MAPE. Therefore, whether the inclusion of dynamic parameter adaptation yields better performance depends on the dataset, and currently no definitive conclusion can be drawn.

## 4.5 Downstream Application: Real-World Ecological Time Series

Table 7: Filtering performance. We measure the MSE in recovering the original observations from doubly noisy data generated with varying levels of synthetic Gaussian noise.

| Noise level ($\epsilon$) | ELTO-KF | ELTO-AKF |
|---|---|---|
| 1 | 0.0126 | **0.0120** |
| 10 | 0.0569 | **0.0419** |
| 100 | 1.3995 | **1.2602** |

We validated ELTO-AKF using the Canadian lynx and snowshoe hare records (Elton & Nicholson, 1942). These records exhibit non-stationary Lotka-Volterra dynamics (Lotka, 1925; Volterra, 1926) and are known to contain outliers and phase mismatches. Lacking ground truth data on the state variables, we followed the procedure in Section 4.4 and treated the original observations as validation targets ($\boldsymbol{Y}^{\text{val}}$). We injected

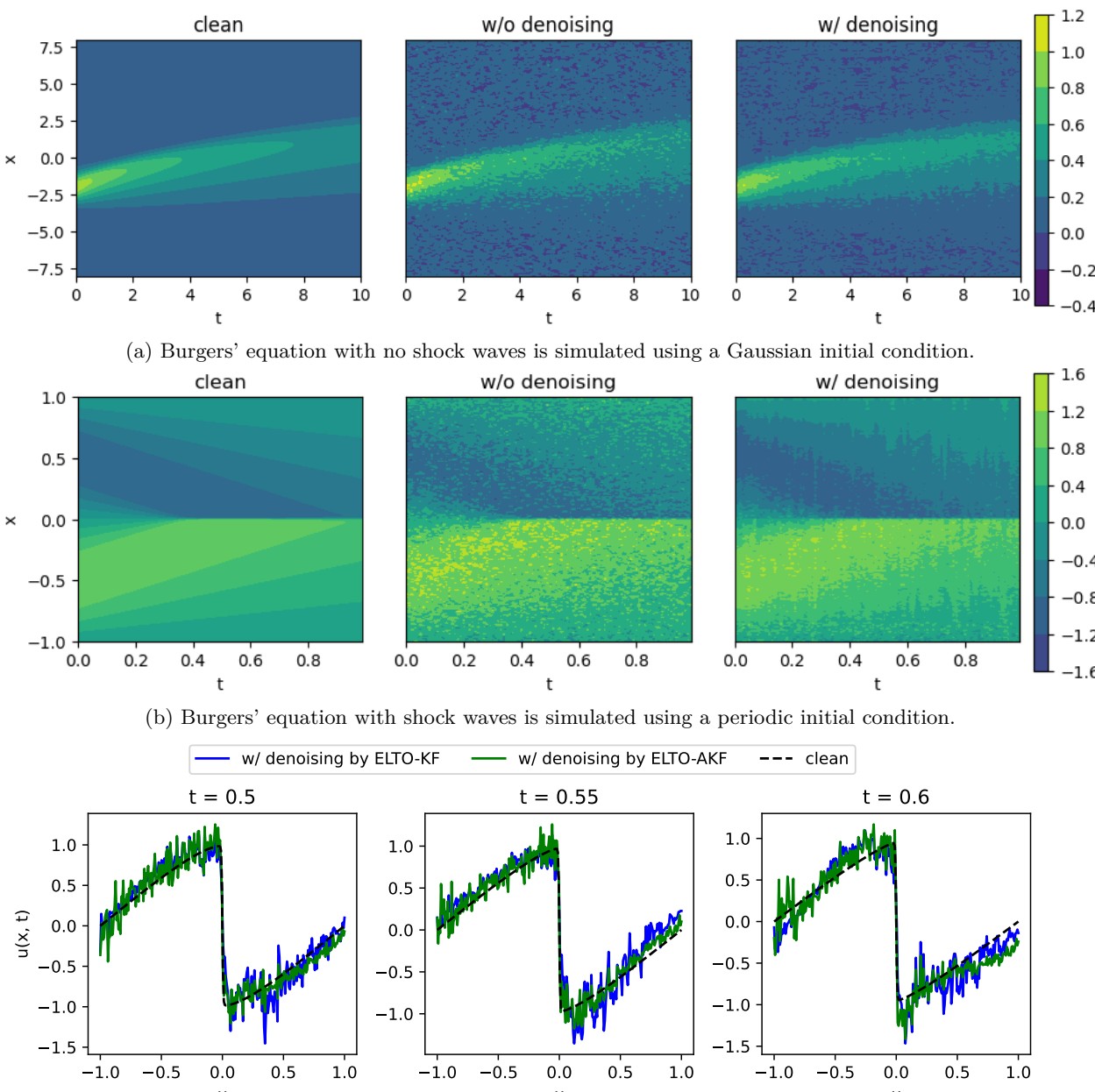

(a) Burgers' equation with no shock waves is simulated using a Gaussian initial condition.

(b) Burgers' equation with shock waves is simulated using a periodic initial condition.

(c) Abrupt change in system dynamics governed by the Burgers' equation caused by shock formation.

Figure 4: The state-variable data are depicted under three conditions: the clean data, the noisy ($\epsilon = 50$) observed data, and the denoised data produced by the ELTO-AKF, which shows the smallest MSE to the clean data.

Table 8: Discovery of the Lotka–Volterra dynamics in the lynx–hare dataset. We compare the discovery process with and without ELTO-AKF smoothing as a denoising step prior to trajectory-matching refinement. Lower MAPE indicates closer agreement with the reference dynamics: $\dot{x}_1 = -0.84x_1 + 0.026x_1x_2$, and $\dot{x}_2 = 0.55x_2 - 0.028x_1x_2$, for which the expected coefficients are statistically derived (Howard, 2009).

| Method | Identified ODEs | | MAPE |
|---|---|---|---|
| w/o ELTO-AKF | $\dot{x}_1 = -0.843x_1 + 0.0266x_1x_2,$ | $\dot{x}_2 = 0.547x_2 - 0.0281x_1x_2$ | 0.907 |
| w/ ELTO-AKF | $\dot{x}_1 = -0.830\,x_1 + 0.0261x_1x_2,$ | $\dot{x}_2 = 0.554x_2 - 0.0282x_1x_2$ | **0.772** |

varying levels of additional synthetic noise, $\frac{\epsilon}{100}\mathcal{N}(0,1)$, into the input data and then optimized the filter to recover the original observations from these doubly noisy inputs, evaluating the robustness of the data-driven Kalman filters. Table 7 demonstrates that ELTO-AKF consistently outperforms ELTO-KF across all noise levels, confirming its ability to isolate the complex dynamics inherent in real-world measurements. Consequently, these filters show strong potential for further denoising the original observations without relying on ground truth supervision.

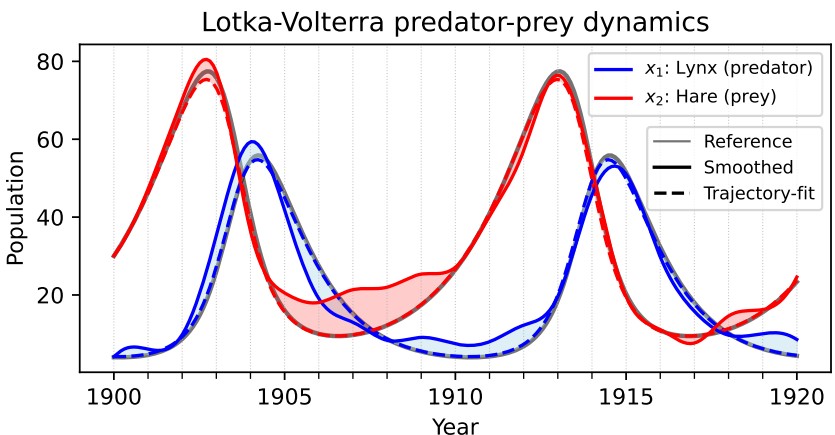

Figure 5: Recovered Lotka–Volterra models from the lynx–hare data. The trajectory-fit solution and the smoothed observations closely match the simulated solution from the reference coefficients.

We also demonstrate that ELTO-AKF with its memory factors ablated can be optimized to smooth observations, thereby facilitating the downstream discovery of the governing system of ODEs. The original 20-point series is upsampled to 200 evenly spaced points using cubic splines to form the input observations $Y$, and we apply a sparse-regression discovery algorithm (Rudy et al., 2017) to recover the underlying Lotka-Volterra dynamics. The discovery process proceeds in three steps: (i) sparse regression over a degree-2 polynomial library identifies the active terms; (ii) iterative Kalman smoothing produces a denoised trajectory; and (iii) a trajectory-matching refinement re-optimizes the coefficient values by integrating the candidate ODE forward in time and minimizing the squared error against the denoised trajectory, thereby bringing the recovered coefficients closer to the true dynamics. Table 8 reports the results of an ablation study comparing the full discovery process against a variant that skips the smoothing and runs the trajectory-matching refinement directly on the upsampled observations. Without the trajectory-matching refinement, the sparse-regression step alone yields a MAPE of 5.41 in both conditions, which is far from the reference coefficients; however, the refinement step, when fed the smoothed trajectory, lowers the MAPE to 0.772. This indicates that ELTO-AKF smoothing yields a clear downstream benefit by providing the trajectory-matching step with a cleaner integration target, as shown in Figure 5.

## 5    Conclusions

**Summary.**   We introduced the ELTO-AKF, which addresses the limitations of the standard ELTO-KF in learning non-stationary dynamics through the novel structured noise adaptation. The computationally tractable SB parameterization is proposed, guaranteeing the symmetric positive-definite property of the noise covariance matrices by updating their Cholesky factors during the dynamic parameter adaptation. This structured parameterization grants our proposed method the capability of structured noise adaptation: learning globally robust, time-invariant noise models through optimization, while dynamically tracking time-varying system dynamics using filter residuals. Empirical results in dynamic state estimation reveal that the ELTO-AKF significantly outperforms baseline methods in challenging non-stationary processes. Furthermore, when applied as a denoising method for data-driven PDE discovery, our ELTO-AKF method reduces

the identification error for complex systems governed by the viscous Burgers' equation, whose solutions exhibit shock waves.

**Limitations and Future Work.** While ELTO-AKF demonstrates significant robustness and circumvents the need for noiseless latent states by optimizing over noisy validation observations, we acknowledge certain limitations in the current parameter fitting procedure. In SSMs, principled gradient-based approaches, such as Maximum Marginal Likelihood Estimation (MMLE) via particle methods, provide an elegant framework for parameter learning without ground truth data. However, because our filtering inference operates entirely via distribution embeddings within the RKHS, it inherently bypasses explicit probability density functions. Consequently, the current RKHS formulation lacks an explicit observation likelihood or a tractable pre-image approximation, rendering likelihood-based methods like MMLE not directly applicable and encouraging other optimization techniques (e.g., derivative-free CMA-ES). Furthermore, our dynamic tracking mechanism relies on predefined memory factors $(\alpha, \beta)$ to govern adaptation rates. Therefore, formulating a structured likelihood approximation within the RKHS to enable principled, gradient-based optimization, alongside exploring fully adaptive strategies for the memory factors, remain important directions for future research.

**Broader Impact Statement.** This work advances sequential Bayesian filtering by introducing a structured parameterization for learning noise covariance matrices. We hope our approach to designing noise-robust covariance matrices benefits other researchers working on non-stationary filtering. While our evaluations focus on domains like LiDAR tracking, robust state estimation is fundamental to general navigation. Therefore, we acknowledge the dual-use potential of this technology, particularly in surveillance applications.

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

# A  Matrix-based Spectral Learning Algorithm for Operator Identification

This section summarizes the spectral learning algorithm proposed by (Ke et al., 2025) for estimating the Embedded Latent Transfer Operators (ELTO). This algorithm provides a constructive method for deriving the latent state estimates directly from the observation history, which enables the computation of the empirical matrices $\boldsymbol{T}$ and $\boldsymbol{O}$ used in the main text.

The empirical estimation process begins with a finite observation sequence $\boldsymbol{Y} = [\boldsymbol{y}_1, \ldots, \boldsymbol{y}_T]$ and a window size $h$. Vectors for the past and future, $\boldsymbol{y}_{p,n}$ and $\boldsymbol{y}_{f,n}$, are constructed for each time step $n = 1, \ldots, N$, where $N := T - 2h + 2$ is the total number of windows:

$$\boldsymbol{y}_{p,n} := [\boldsymbol{y}_{n+h-2}, ..., \boldsymbol{y}_{n-1}]^\top, \quad \boldsymbol{y}_{f,n} := [\boldsymbol{y}_{n+h-1}, ..., \boldsymbol{y}_{2h-2+n}]^\top. \tag{19}$$

These vectors are then mapped into the RKHS. For example, for the past window, this results in a feature matrix:

$$\Phi_{p,n} := [\phi_y(\boldsymbol{y}_{n+h-2}), \ldots, \phi_y(\boldsymbol{y}_{n-1})]. \tag{20}$$

The empirical covariance matrices are then computed from the features of these windows. Let $\Phi_p = [\Phi_{p,1}, \ldots, \Phi_{p,N}]$ and $\Phi_f = [\Phi_{f,1}, \ldots, \Phi_{f,N}]$ be the feature matrices for the entire sequence of past and future windows, and let $\boldsymbol{Q}_N := \boldsymbol{I}_N - \frac{1}{N}\boldsymbol{1}_N\boldsymbol{1}_N^\top$ be a centering matrix. The covariance matrices are given by:

$$\boldsymbol{C}_{pp} := \frac{1}{N}\Phi_p\boldsymbol{Q}_N\Phi_p^\top, \quad \boldsymbol{C}_{ff} := \frac{1}{N}\Phi_f\boldsymbol{Q}_N\Phi_f^\top, \quad \boldsymbol{C}_{fp} := \frac{1}{N}\Phi_f\boldsymbol{Q}_N\Phi_p^\top. \tag{21}$$

After computing the Cholesky factorizations $\boldsymbol{C}_{ff} = \boldsymbol{L}\boldsymbol{L}^\top$ and $\boldsymbol{C}_{pp} = \boldsymbol{M}\boldsymbol{M}^\top$, a singular-value decomposition (SVD) of the normalized cross-covariance is performed:

$$\boldsymbol{L}^{-1}\boldsymbol{C}_{fp}(\boldsymbol{M}^{-1})^\top \approx \hat{\boldsymbol{U}}\hat{\boldsymbol{S}}\hat{\boldsymbol{V}}^\top. \tag{22}$$

This yields a projection matrix $\boldsymbol{B} := \hat{\boldsymbol{S}}^{1/2}\hat{\boldsymbol{V}}^\top\boldsymbol{M}^{-1}$. To construct the latent state, a feature matrix $\Phi_S = [\phi_y(\boldsymbol{y}_{e_1}), \ldots, \phi_y(\boldsymbol{y}_{e_{|S|}})]$ is formed from a subset of observations $S \subseteq \{\boldsymbol{y}_0, \ldots, \boldsymbol{y}_T\}$. A vector of weights, $w$, is then learned by optimizing a loss function related to the canonical correlations. The latent state vector is then constructed as:

$$\boldsymbol{x}_n = \boldsymbol{B}\Phi_{p,n}^\top\Phi_S w, \tag{23}$$

where $\Phi_S w$ represents a projection into the RKHS learned via the spectral method to maximize canonical correlations.

With the latent state sequence $\{\boldsymbol{x}_n\}$ obtained, the system's dynamics are modeled. Using the feature matrices defined previously, the embedded operators $\mathcal{T}_e$ and $\mathcal{O}_e$ are empirically estimated as regularized operators:

$$\hat{\mathcal{T}}_e = \boldsymbol{\Psi}_2\boldsymbol{\Psi}_1^\top(\boldsymbol{\Psi}_1\boldsymbol{\Psi}_1^\top + \epsilon_t\mathcal{I})^{-1}, \tag{24}$$

$$\hat{\mathcal{O}}_e = \boldsymbol{\Phi}\boldsymbol{\Psi}^\top(\boldsymbol{\Psi}\boldsymbol{\Psi}^\top + \epsilon_o\mathcal{I})^{-1}. \tag{25}$$

where $\boldsymbol{\Psi}_1 := \boldsymbol{\Psi}_{:,1:N-1}$, $\boldsymbol{\Psi}_2 := \boldsymbol{\Psi}_{:,2:N}$, and $\epsilon_t, \epsilon_o > 0$. For the matrix-based implementation of the Kalman filter used in the main text (Section 3.2), we require the $N \times N$ matrix representations of these operators, which are derived by applying the kernel trick. Following (Gebhardt et al., 2019), we first define the necessary Gram matrices:

$$\boldsymbol{G}_{\tilde{x}} = \boldsymbol{\Psi}_1^\top\boldsymbol{\Psi}_1, \quad \boldsymbol{G}_{\tilde{x}x} = \boldsymbol{\Psi}_1^\top\boldsymbol{\Psi}_2, \quad \boldsymbol{G}_x = \boldsymbol{\Psi}^\top\boldsymbol{\Psi}, \quad \boldsymbol{G}_{yx} = \boldsymbol{\Phi}^\top\boldsymbol{\Psi}. \tag{26}$$

This allows the operator in Equation 24-25 to be re-expressed as the $N \times N$ matrices $\boldsymbol{T}$ and $\boldsymbol{O}$:

$$\boldsymbol{T} = (\boldsymbol{G}_{\tilde{x}} + \epsilon_t\boldsymbol{I}_N)^{-1}\boldsymbol{G}_{\tilde{x}x}, \tag{27}$$

$$\boldsymbol{O} = (\boldsymbol{G}_x + \epsilon_o\boldsymbol{I}_N)^{-1}\boldsymbol{G}_{yx}^\top. \tag{28}$$

These matrices $\boldsymbol{T}$ and $\boldsymbol{O}$, along with the noise matrices $\boldsymbol{Q}$ and $\boldsymbol{R}$, form the basis of the Kalman filtering process described in Section 3.2. The initialization of sequential state estimation with ELTOs is then given in Algorithm 3.

---

**Algorithm 3** Learning system dynamics with the Embedded Latent Transfer Operator (ELTO)

---

1: **Input:** $Y = [y_1, \ldots, y_T]$, kernel function $k(\cdot, \cdot)$ and regularization parameters $\epsilon_t$, $\epsilon_o$.
2: Compute $\boldsymbol{\Phi}, \boldsymbol{\Psi}$ following Equation 2 using $Y$
3: Compute $\boldsymbol{T} = (\boldsymbol{G}_{\tilde{x}} + \epsilon_t \boldsymbol{I}_N)^{-1} \boldsymbol{G}_{\tilde{x}x}, \quad \boldsymbol{O} = (\boldsymbol{G}_x + \epsilon_o \boldsymbol{I}_N)^{-1} \boldsymbol{G}_{yx}^{\top}$ following Equation 26
4: Sample $N$ basis vectors $\boldsymbol{u}_1, \ldots, \boldsymbol{u}_N$ from a multivariate Uniform distribution $\mathcal{U}(0, 1)$
5: Define the kernel matrix $\boldsymbol{K}_0$ where $(\boldsymbol{K}_0)_{i,j} = k_x(\boldsymbol{u}_i, \boldsymbol{x}_j)$ for $i = 1, \ldots, N$ and $j = 1, \ldots, N$
6: Compute the initial state embedding $\boldsymbol{C}_0 = (\boldsymbol{G}_x + \epsilon_o \boldsymbol{I}_N)^{-1} \boldsymbol{K}_0$
7: Compute mean $\boldsymbol{m}_0$ and variance $\boldsymbol{S}_0$ over the columns of $\boldsymbol{C}_0$
8: **Output:** $\boldsymbol{T}, \boldsymbol{O}, \boldsymbol{m}_0$, and $\boldsymbol{S}_0$

---

# B  Proof of Distance Monotonicity in RKHS

The following proof validates a general and crucial property of the radial basis function (RBF) kernel. We demonstrate that for any two arbitrary points $\mathbf{y}, \mathbf{y}' \in \mathbb{Y}$, the RBF kernel $k(\mathbf{y}, \mathbf{y}') = \exp(-\gamma||\mathbf{y} - \mathbf{y}'||_{\mathbb{Y}}^2)$, which is a shift-invariant kernel, guarantees a monotonic relationship between the input-space distance and the feature-space distance. This specific connection is a foundational result in kernel methods (Schölkopf, 2000), forming the basis of what is often termed the kernel distance (Phillips & Venkatasubramanian, 2011).

**Proposition 2.** For any two arbitrary points $\mathbf{y}, \mathbf{y}' \in \mathbb{Y}$ from the observation space, let the feature map $\phi : \mathbb{Y} \to \mathbb{H}$ be induced by the RBF kernel $(k(\mathbf{y}, \mathbf{y}') = \exp(-\gamma||\mathbf{y} - \mathbf{y}'||_{\mathbb{Y}}^2), \gamma > 0)$. The squared Euclidean distance in the feature space, $||\phi(\mathbf{y}) - \phi(\mathbf{y}')||_{\mathbb{H}}^2$, is a monotonically increasing function of the squared Euclidean distance in the observation space, $||\mathbf{y} - \mathbf{y}'||_{\mathbb{Y}}^2$.

**Proof.** We define the squared Euclidean distance in the Reproducing Kernel Hilbert Space (RKHS) $\mathbb{H}$ using the inner product $\langle \cdot, \cdot \rangle_{\mathbb{H}}$:

$$||\phi(\mathbf{y}) - \phi(\mathbf{y}')||_{\mathbb{H}}^2 = \langle \phi(\mathbf{y}) - \phi(\mathbf{y}'), \phi(\mathbf{y}) - \phi(\mathbf{y}') \rangle_{\mathbb{H}} \tag{29}$$

$$= \langle \phi(\mathbf{y}), \phi(\mathbf{y}) \rangle_{\mathbb{H}} + \langle \phi(\mathbf{y}'), \phi(\mathbf{y}') \rangle_{\mathbb{H}} - 2\langle \phi(\mathbf{y}), \phi(\mathbf{y}') \rangle_{\mathbb{H}}. \tag{30}$$

By the kernel trick, the inner product in the feature space can be computed by the kernel function in the input space:

$$\langle \phi(\mathbf{y}), \phi(\mathbf{y}') \rangle_{\mathbb{H}} = k(\mathbf{y}, \mathbf{y}'). \tag{31}$$

Substituting this into Equation 30, we get:

$$||\phi(\mathbf{y}) - \phi(\mathbf{y}')||_{\mathbb{H}}^2 = k(\mathbf{y}, \mathbf{y}) + k(\mathbf{y}', \mathbf{y}') - 2k(\mathbf{y}, \mathbf{y}'). \tag{32}$$

Specifically, the RBF kernel is shift-invariant (stationary), and $k(\mathbf{y}, \mathbf{y}) = k(\mathbf{y}', \mathbf{y}') = \exp(0) = 1$:

$$||\phi(\mathbf{y}) - \phi(\mathbf{y}')||_{\mathbb{H}}^2 = 2\left(1 - \exp(-\gamma||\mathbf{y} - \mathbf{y}'||_{\mathbb{Y}}^2)\right). \tag{33}$$

Since $||\mathbf{y} - \mathbf{y}'||_{\mathbb{Y}}^2 \geq 0$ and $\gamma > 0$, as $d$ increases, the entire $||\phi(\mathbf{y}) - \phi(\mathbf{y}')||_{\mathbb{H}}^2$ increases. This directly proves that the feature-space squared distance is a monotonically increasing function of the input-space squared distance, confirming the property. $\square$

The significance of this general proof is that it provides the formal justification for kernel-based optimization frameworks. By proving this monotonic property, we confirm that minimizing the squared residual in the RKHS (i.e., $||\phi(\mathbf{y}) - \phi(\mathbf{y}')||_{\mathbb{H}}^2$) is equivalent to minimizing the squared residual in the original observation space (i.e., $||\mathbf{y} - \mathbf{y}'||_{\mathbb{Y}}^2$). This ensures that an optimal solution found in the RKHS also corresponds to an optimal solution in the observation space (i.e., Equation 16).

## C  Extended Ablation and Analysis

### C.1  Core Architectural Ablation

**Necessity of the Scalar-Block (SB) Structure.**  To evaluate the SB parameterization, we compare ELTO-AKF against an ablation baseline lacking the SB structure. The full model (Algorithm 2) projects noise estimates onto the SB space and updates its low-dimensional Cholesky factors ($\boldsymbol{M}_k, \boldsymbol{L}_k$) to guarantee positive definiteness. The ablation skips this projection, updating the dense $\boldsymbol{R}$ and $\boldsymbol{Q}$ matrices directly to simulate a standard AEKF. Both models are evaluated on non-stationary trajectories (Section 4.2) with $\alpha = \beta = 0.9$ and initialized from the same data-driven $\boldsymbol{Q}_0, \boldsymbol{R}_0$ (detailed in footnote 7).

Table 9: MSE ($\times 10^{-2}$) of structured vs. unstructured adaptive noise models on non-stationary trajectories ($\alpha = \beta = 0.9$).

| Noise Level | ELTO-AKF | |
|---|---|---|
| SB structure | With | Without |
| Default | **8.8112** | 26.7737 |
| High | **11.2288** | 20.1178 |
| Very High | **35.0613** | 44.8016 |

As shown in Table 9, ELTO-AKF outperforms the unstructured ablation. Direct updates to dense matrices in the ablation model fail to preserve the symmetric positive-definite (SPD) property, triggering numerical instability and elevated tracking errors. Furthermore, the robustness against varying and extreme noise conditions is provided in Table 3 in the main text.

### C.2  Hyperparameter Sensitivity

**Memory Factors ($\alpha, \beta$).**  Table 10 shows the sensitivity of the adaptation rates under both matched (train noise equals test noise) and mismatched conditions.

Table 10: MSE under varying adaptation rates ($\alpha = \beta$). Results are grouped by matched (MSE $\times 10^{-2}$) and mismatched (MSE $\times 10^{-1}$) noise conditions.

| Noise Level | | Adaptation Rates ($\alpha = \beta$) | | | |
|---|---|---|---|---|---|
| Train | Test | 0.9 | 0.7 | 0.5 | 0.3 |
| *Matched Conditions (Train = Test)* | | | | | |
| Default | Default | **8.8112** | 10.7419 | 10.9886 | 29.4613 |
| High | High | **11.2288** | 11.4485 | 13.0108 | 14.2895 |
| Very High | Very High | **35.0613** | 35.8572 | 41.6618 | 41.0659 |
| *Mismatched Conditions (Train $\neq$ Test)* | | | | | |
| Default | High | **4.3350** | 4.4122 | 5.2780 | 6.0676 |
| Default | Very High | **4.9642** | 6.0275 | 6.6339 | 7.1362 |

**Scalar-Block Size ($k$).**  Table 11 shows the impact of the scalar-block number $k$. Increasing $k$ can improve expressiveness, and $k = 10$ achieves the lowest MSE in this experiment. However, the relationship is not strictly monotonic, likely because larger parameter spaces can make CMA-ES optimization more difficult. We use $k = 5$ as a practical default balancing accuracy and optimization cost.

Table 11: Sensitivity to scalar-block number ($k$).

| Model | ELTO-KF | ELTO-AKF | | |
|---|---|---|---|---|
| **Scalar-block num** ($k$) | - | $k = 2$ | $k = 5$ | $k = 10$ |
| **MSE** | 0.1600 | 0.1043 | 0.1133 | **0.0694** |

**Base ELTO Parameters ($h$ and kernel functions).** Tables 12 and 13 evaluate the foundational operator hyperparameters. We evaluate our method using three standard kernels: an L2-distance Laplacian, Matérn ($\nu = 3/2$), and RBF. The scale parameter $\gamma$ for each is uniformly defined as $1/d$, with $d$ being the number of features. Regarding the window size, ELTO-AKF maintains stable performance across different $h$ values, consistently outperforming the non-adaptive ELTO-KF baseline. As for the kernel functions, ELTO-AKF substantially improves over ELTO-KF for the Matérn and RBF kernels, while the Laplacian kernel is less suitable in this setting. These results suggest that structured adaptation is not solely tied to the RBF kernel, although kernel choice remains important. Ultimately, we select $h = 5$ and the RBF kernel to balance temporal smoothing and nonlinear representation.

Table 12: Sensitivity to historical window size ($h$).

| Model | ELTO-KF | | | ELTO-AKF | | |
|---|---|---|---|---|---|---|
| **Window size ($h$)** | 5 | 10 | 20 | 5 | 10 | 20 |
| **MSE** | 0.2426 | 0.2412 | 0.2415 | **0.1113** | 0.1296 | 0.1310 |

Table 13: Sensitivity to different kernel functions.

| Model | ELTO-KF | | | ELTO-AKF | | |
|---|---|---|---|---|---|---|
| **Kernel** | Laplacian | Matérn | RBF | Laplacian | Matérn | RBF |
| **MSE** | 0.2422 | **0.2418** | 0.2494 | 0.3168 | 0.1544 | **0.1513** |

### C.3 Computational Complexity and Empirical Runtime

Table 14 evaluates the computational cost of ELTO-AKF, which comprises operator training, CMA-ES optimization, and inference. While operator construction involves Gram matrix computation and inversion that scale polynomially with the number of training samples $N$, this process is computed offline and, crucially, is completely independent of the scalar-block size $k$. During inference, the state and covariance updates require matrix inversions. As $k$ increases, CMA-ES optimization time grows because estimating the symmetric matrices $\boldsymbol{Q}_\theta$ and $\boldsymbol{R}_\theta$ expands the search space to $\mathcal{O}(k^2)$. Crucially, while a larger $k$ increases the parameter optimization cost, both operator training and inference times remain completely unaffected. Note that the high variance in the baseline ELTO-KF's optimization time stems from frequent SVD failures caused by its lack of structural SPD guarantees during the unconstrained search.

Table 14: Computational complexity analysis of ELTO-AKF

| Model | ELTO-KF | ELTO-AKF | | |
|---|---|---|---|---|
| Scalar-block size ($k$) | - | 2 | 5 | 10 |
| Operator Training (s) | $3.82 \pm 0.18$ | $3.78 \pm 0.02$ | $3.80 \pm 0.15$ | $3.81 \pm 0.03$ |
| CMA-ES Optimization (s) | $137.36 \pm 45.27$ | $282.91 \pm 0.96$ | $395.58 \pm 1.11$ | $507.92 \pm 0.71$ |
| Inference (s) | $1.40 \pm 0.01$ | $2.778 \pm 0.007$ | $2.782 \pm 0.006$ | $2.773 \pm 0.008$ |

