# OpenReview forum: "Structured Noise Adaptation for Sequential Bayesian Filtering with Embedded Latent Transfer Operators"
_TMLR — Accepted by TMLR_

### Review · Reviewer_SYCW · 2026-02-11

**Summary Of Contributions:**

The work proposes ELTO-AKF, an extension of ELTO-based Kalman filter to handle robustness in non-stationary processes. This follows previous line of works that consider Kalman filters defined on RKHSs with data-driven dynamics/observation models learned by ELTO. The main contributions can be summarised as proposing:
- a training-based noise covariance estimation method with an imposed sparsity structure
- a method for dynamically adapting noise covariance matrices

The proposed method outperforms a standard ELTO-based Kalman filter on various filtering and denoising tasks under the non-stationary regime.

**Audience:**

No

**Audience Explanation:**

The paper tackles the challenging problem of filtering under dynamically evolving noise covariances. However, I do not believe the paper in the current form addresses this problem adequately and convincingly, as explained above. If the authors are able to address some of these issues, I would be happy to reconsider my evaluation.

**Broader Impact Concerns:**

There are no negative ethical implications of the work as far as I can see.

**Claims And Evidence:**

No

**Claims Explanation:**

Overall, I find that the paper suffers from major clarity issues, which makes it challenging to understand and be convinced by some of the modelling choices made  (see __Requested Changes__ below for more details). In particular, I am unsure about how reasonable it is to assume access to the ground truth measurements $Y^{true}$ to do the training in Algorithm 1 without directly knowing the dynamics and observation models to generate synthetic data from. If this is the case, I struggle to see how an ELTO-based filter is useful over a standard Kalman-type filter as, from my understanding, the purpose of the ELTO-based filter is to perform filtering while simultaneously inferring the dynamics/observation operator from data. If there are other reasons for using ELTO-based filter, a comparison with a standard Kalman-type filter (e.g. the adaptive extended Kalman filtering, cited by the authors) is missing to demonstrate its benefits.

Aside from this, there are other issues regarding clarity such as the baseline model details, and what hyperparameters are being tuned in §4.2, which makes it hard to interpret these results. In addition, a discussion about how the method performs in the challenging+realistic scenario when there is a distribution shift at test time would be desirable as I believe this is where we would expect the adaptive covariance learning method to be useful. However, from what I can see, such a test is only demonstrated briefly in Table 7 in the Appendix.

For these reasons, while I have checked that the theoretical result in Appendix B is accurate, the overall motivational and clarity issues leave me unconvinced overall about the claims in the submission.

**Requested Changes:**

Following from my above comments regarding clarity,  I would appreciate if the authors could please elaborate on the following points:
- I believe details on the estimation method for the noise matrices $Q$ and $R$ in the standard ELTO-based Kalman filter would be helpful to include for clarity purpose. The authors only write "However, standard ELTO-based approaches typically derive a static/non-adaptive process noise matrix $Q$ from the residuals of the learned operators" (page 5). I believe expanding on this is important as it would further highlight the difference with the proposed approach. In my understanding, a key difference is that the previous ELTO approach does not learn $Q$ and $R$ based on mismatch of the filter predictions with a "ground truth" data $Y^{truth}$, whereas the proposed method does, which makes it more robust. However, this leads to my next question.
- How reasonable is it to assume that the ground truth $Y^{true}$ is available in practice (in Algorithm 1)? The only situation I can think of where this is possible is when one has access to the models generating the states and observations to synthetically generate these ground truths. However, in this case, we can just use a standard off-the-shelf filter. Since part of the appeal of the ELTO-based approach seems to be in learning these models directly from data (data-driven identification), it seems to be directly contradicting its purpose?
- The dynamical adaptation methodology is not well-explained, only referring to parallels with [Akhlaghi et al. (2017)]. It would help if  the authors could motivate better why the update choice in (18) was considered, e.g. an intuitive reasoning why this is able to successfully "adapt" to the non-stationarity of the process.
- The notation $\Upsilon_x$ is not introduced in equations (5) and (6).
- Insufficient details regarding the baselines SubKKR, KKR and ELTO-KF in the noisy pendulum experiment. Just by reading the background, I am not even sure what the differences are between KKR and ELTO-KF are, let alone SubKKR, which does not seem to be introduced anywhere.
- I don't understand what the authors refer to as "dataset-specific tuning" in the experiments in Section 4.2, and what are these "hyperparameters" that the authors are referring to? Is this referring to the kernel hyperparameters such as the lengthscale? If so, how is this learned in the first batch of experiments (left columns) in Table 2?
- Demonstration of the method when there is a distribution shift at test-time (i.e. different noise characteristics during training vs test) would be useful in my opinion to show the benefits of adaptive covariance learning (Algorithm 2). If there is no distribution shift, then it seems to me that Algorithm 1 suffices.

---

> ### Author Response · Authors · 2026-05-01
>
> We sincerely thank you for the constructive feedback. We agree that if noiseless ground truth or underlying models were available, standard model-based filters would indeed be sufficient. We apologize for the misunderstanding caused by our misleading notation in the initial manuscript. We are also grateful for the suggestion to include AEKF experiments, which has significantly strengthened the empirical evidence of our work. Below, we address the requested changes point by point.
>
> > Regarding the availability of ground truth:
>
> We thank the reviewer for pointing this out. We sincerely apologize for the misleading notation $\boldsymbol{Y}^{\textrm{true}}$ in Equation 16 and Algorithm 1. To clarify, our method does not assume access to unobserved noiseless states or the underlying dynamics models. This term refers to noisy observations from a held-out validation set (corrected to $\{\boldsymbol{Y}^{\textrm{val}}_{t}\}$ in the revision). We optimize noise covariances (hyperparameter tuning) via CMA-ES on this validation set, which is strictly separated from the training data used to learn the operators $\boldsymbol{T}$ and $\boldsymbol{O}$, ensuring there is no data leakage. We note that using validation observations to tune the parameters of Kalman filters is a common practice in the literature, which we have adopted. The main difference between ELTO-based and traditional filters is that we initialize the transition and observation matrices directly from training data, rather than explicitly defining and subsequently tuning them, as is typically required for model-based filters.
>
> >Regarding the comparison with standard Kalman filters (AEKF):
>
> We appreciate your suggestion on additional experiments. As shown in the revised Table 1, we have added model-based AEKFs (with constant-velocity and linear priors) for comparison to further demonstrate the benefits of the ELTO-based approach. In the default-noise setting, the model-based AEKF (CV) achieves the lowest error, since its priors are well aligned with the relatively simple dynamics. However, as the process or observation noise increases, the AEKF baselines degrade significantly, while ELTO-AKF retains low error and outperforms both AEKF variants under high process noise and high observation noise. This supports our claim that the data-driven, structured-noise design of ELTO-AKF is most beneficial especially in the regimes where fixed analytical priors break down. We also added Table 1(b) for a clearer conceptual comparison across all baselines.
>
> > Regarding the estimation for noise matrices:
>
> As suggested, we have expanded these details (Section 3.2 in the manuscript). In the standard ELTO-KF, the noise matrices $\boldsymbol{Q}$ and $\boldsymbol{R}$ are static and defined as scaled identity matrices (e.g., $\boldsymbol{Q}=\epsilon_{q} \boldsymbol{I},~ \boldsymbol{R}=\epsilon_{r} \boldsymbol{I}$) without further updates. In contrast, our proposed method learns and updates parameterized covariance matrices ($\boldsymbol{Q}_ {\theta}$ and $ \boldsymbol{R}_ {\theta}$) to better capture dynamical changes in non-stationary processes.
>
> >Regarding the dynamical adaptation methodology:
>
> Following your suggestion, we have added intuitive reasoning for this update step (Section 3.3 in the revised manuscript). The memory factors ($\alpha$ and $\beta$) balance the covariance updates by weighing recent innovations against past covariance estimates, thereby governing the rate of adaptation and how the filter reacts to transient noise. To successfully adapt to a non-stationary process, the AEKF tracks the filter's innovations and posterior residuals to detect changes in system dynamics. More specifically, a large innovation signals a shift or mismatch in the system dynamics, whereas a large posterior residual indicates that the updated state still struggles to reconcile with (possibly unreliable) incoming sensor data. As illustrated in the new Figure 4(c), these AEKF updates allow ELTO-AKF to successfully capture abrupt dynamic changes due to shock waves in Burgers' equation.
>
> >Regarding the notations:
>
> We apologize for the oversight in our notation. We have corrected the typos, unified the notation to $\boldsymbol{\Psi}$, which is defined in Eq.2 in the revised manuscript.
>
> >Regarding baseline details:
>
> In Section 4.1, we have added a paragraph and Table 1(b) to further explain the previous works used as baselines and clarifying their mechanisms.

---

> ### Author Response · Authors · 2026-05-01
>
> >Regarding dataset-specific tuning and hyperparameters:
>
> In Section 4.2, the term "hyperparameters'' refers specifically to the initial noise covariance scalars ($\epsilon_q, \epsilon_r$) optimized via CMA-ES, rather than kernel parameters such as lengthscales. We have clarified this distinction in the revised text. Furthermore, Table 2 now contrasts two updated regimes: a "Full Optimization'' regime (previously termed "Optimized''), where hyperparameters are tuned via CMA-ES for a specific dataset, and a "Pure Adaptation'' regime (previously termed "Adaptive''), where they are adjusted solely through online adaptation without dataset-specific tuning. We set $\alpha = \beta = 1$ and $\alpha = \beta = 0.9$ for the Full Optimization and Pure Adaptation, settings respectively.
>
> >Regarding performance against distribution shifts:
>
> We have added a new experiment (Table 3 in the revised manuscript) to specifically evaluate scenarios with a clear distribution shift between training and testing. The results demonstrate that ELTO-AKF maintains its robustness under these train–test distribution shifts better than the ELTO-KF baseline.

---

### Review · Reviewer_YozM · 2026-03-08

**Summary Of Contributions:**

Summary:
The paper addresses a key limitation in standard Embedded Latent Transfer Operator (ELTO) Kalman filters: their reliance on fixed, non-adaptive noise covariances that struggle with non-stationary processes. The authors introduce a novel filtering method, ELTO-AKF, which incorporates a structured noise model adaptation.

Key Contributions:
The authors propose a Scalar-Block (SB) parameterization to make the learning of noise covariance matrices computationally tractable. They ensure the Symmetric Positive-Definite (SPD) property of these matrices is maintained during adaptation by utilizing a Cholesky parameterization. The approach successfully couples the learning of a globally robust, time-invariant noise model (using derivative-free CMA-ES optimization) with dynamic parameter adaptation that responds to local changes in the environment. Empirical validations span multiple domains, including a noisy pendulum simulation, non-stationary Lidar tracking, and a downstream task of data-driven partial differential equation (PDE) discovery (Burgers' equation).

Strengths:
The theoretical foundation is sound, particularly the proof guaranteeing the SPD property of the structured matrices. The ablation study effectively justifies the architectural choices, clearly demonstrating that unstructured adaptation leads to numerical instability and degraded performance. Applying the filtering method to improve data quality for downstream PDE discovery is a highly compelling and practical use case.

Weaknesses:
The method relies on memory factors (alpha and beta) for dynamic adaptation, which are currently fixed based on empirical guidelines or manual tuning. The computational overhead introduced by the CMA-ES optimization and periodic Cholesky updates during the filtering loop is not extensively discussed.

**Audience:**

Yes

**Audience Explanation:**

The TMLR audience encompasses researchers working on state estimation, dynamical systems, and robust machine learning. The integration of Reproducing Kernel Hilbert Space (RKHS) embeddings with adaptive Kalman filtering bridges a significant gap between non-parametric operator learning and classical control theory. Furthermore, the application of this filtering technique to denoise data for the sparse identification of governing PDEs will attract readers interested in scientific machine learning and physics-informed AI.

**Broader Impact Concerns:**

The paper addresses fundamental algorithmic improvements in sequential Bayesian filtering. It does not present immediate negative ethical implications. However, because robust state estimation is a core component of tracking, navigation, and Lidar applications, it inherently dual-uses technology applicable to autonomous systems and surveillance. A brief, standard acknowledgement of the dual-use nature of tracking algorithms in a Broader Impact Statement would suffice, though it is not strictly mandatory given the theoretical focus of the work.

**Claims And Evidence:**

Yes

**Claims Explanation:**

The claims are well-supported through both theoretical and empirical means. The authors provide a formal mathematical proof establishing that their Scalar-Block structure maintains the necessary SPD property for covariance matrices. Empirically, the ELTO-AKF demonstrates superior performance against benchmark methods (like KKR, SubKKR, and standard ELTO-KF) in environments with high process and observation noise. Additionally, the ablation study provides convincing evidence that removing the proposed SB structure results in severe performance drops, thereby validating the necessity of their specific parameterization.

**Requested Changes:**

Comparison with Neural-Aided KFs: The paper claims superior performance in non-stationary and high-noise environments. However, it lacks a comparison with recent state-of-the-art hybrid methods that combine nonlinear filtering with data-driven neural components. A comparison is necessary to determine if the RKHS-based operator approach offers tangible advantages (e.g., stability, interpretability, or data efficiency) over existing neural-augmented sigma-point/cubature frameworks, for examples:
Loo et al., "Sigma-Point Kalman Filter With Nonlinear Unknown Input Estimation via Optimization and Data-Driven Approach for Dynamic Systems," IEEE Trans. Syst., Man, Cybern., Syst., vol. 54, no. 10, pp. 6068-6081, 2024.
Hu et al., "CKFNet: Neural Network Aided Cubature Kalman Filtering," IEEE Signal Process. Lett., vol. 32, pp. 3455-3459, 2025.

Ablation Study on Kernel Selection: The current manuscript relies exclusively on the Radial Basis Function (RBF) kernel. While Appendix B provides a strong theoretical justification for the RBF kernel's monotonic properties, the empirical results lack a comparison with alternative kernel functions (e.g., Matern or Polynomial kernels). Since the performance of RKHS-based methods is highly sensitive to the chosen feature map, the authors must provide a comparative analysis on at least one benchmark task (such as the Noisy Pendulum) using an alternative kernel to demonstrate that the ELTO-AKF's benefits are not strictly an artifact of the RBF kernel's specific geometry

Sensitivity Analysis of Scalar-Block (SB) Dimensions: The authors specify a diagonal parameterization with $k_q = k_r = 5$ blocks for the pendulum task. There is currently no evidence showing how the choice of block size $k$ affects the trade-off between filter accuracy and computational efficiency. A sensitivity study varying the number of sub-blocks in the SB structure is required to justify the "tractable learning" claim and to provide guidance on selecting this parameter for new domains.

Embedding Window Size Impact: The spectral learning algorithm utilizes a window size $h$ to construct past and future feature matrices. While the authors use a window size of 5 for the pendulum task , the manuscript would be strengthened by a brief discussion or plot showing how the quality of the Embedded Latent Transfer Operator (ELTO) identification varies with different window lengths, particularly in the non-stationary Lidar experiment.

Computational Complexity Analysis: Please include a brief theoretical or empirical analysis of the computational overhead introduced by the ELTO-AKF. Specifically, compare the runtime of the dynamic Cholesky decomposition updates and the CMA-ES optimization phase against the baseline ELTO-KF.

Discussion on Memory Factors: The dynamic adaptation heavily relies on the memory factors. Please add a discussion on the sensitivity of the model to these hyperparameters and whether there are potential avenues for making these factors dynamically adaptive rather than statically assigned.

Visual Aids for Broader Audience: While Figure 2 is excellent for detailing the ELTO-AKF specific loop, readers less familiar with the transition from standard state-space models to RKHS might benefit from a comparative diagram.  Consider adding a side-by-side visual comparing the traditional Kalman filter steps in the original observation space with your operator-based steps in the RKHS.

Clarification in Table 2: The distinction between the "Optimized" and "Adaptive" columns is slightly dense. Consider adding a sentence in the main text clearly defining why the adaptive version without dataset-specific tuning still holds value despite higher MSEs.

---

> ### Author Response · Authors · 2026-05-01
>
> We sincerely thank you for the detailed feedback. Based on your comprehensive suggestions, we have added neural-aided baselines and extensive ablation studies, and included more detailed discussions and visualizations to improve clarity. Below, we address your recommendations and detail the corresponding revisions we have made.
>
> >Comparison with Neural-Aided KFs:
>
> We sincerely thank the reviewer for directing us to these highly relevant neural-aided filters, and fully agree that comparing with these hybrid methods is necessary. Accordingly, we did experiments on those two paper we cited in the revised manuscript. Under standard settings, these methods show comparable performance. However, under our specific extreme-noise scenarios, CKFNet encountered numerical instability. This highlights the stability of our RKHS-based operators under extreme noise limits.
>
> **CKF/SPKF experiments**
>
> | Noise Setting           | CKFNet                  | SPKF    | **ELTO-AKF**     |
> |-------------------------|-------------------------|---------|------------------|
> | Default noise           | 8.161                   | 1.6099  | **0.1012**       |
> | High process noise      | $>10^{3}$ (divergence)  | 3.473   | **0.1123**       |
> | High observation noise  | $>10^{4}$ (divergence)  | 12.218  | **8.881**        |
>
> > Kernel Selection:
>
> We conducted an ablation study on kernel selection (Table 13 in the revised appendix). While the baseline ELTO-KF performs marginally better with the Matérn kernel, our proposed ELTO-AKF achieves its best performance using the RBF kernel. We have included these results in the revised appendix.
>
> **Kernel Selection**
>
> | Model      | Laplacian | Matérn      | RBF        |
> |------------|-----------|-------------|------------|
> | ELTO-KF    | 0.2422    | **0.2418**  | 0.2494     |
> | ELTO-AKF   | 0.3168    | 0.1544      | **0.1513** |
>
> >Sensitivity Analysis of Scalar-Block (SB) Dimensions:
>
> We evaluated the effect of the block size $k$ for the SB structure, as detailed in Table 11 in Appendix C. Increasing $k$ can improve expressiveness, and $k=10$ achieves the lowest MSE in this experiment. However, the relationship is not strictly monotonic, likely because larger parameter spaces can make CMA-ES optimization more difficult. We use $k=5$ as a practical default balancing accuracy and optimization cost. Note that Table 9 in the revised manuscript presents an ablation study demonstrating that ELTO-AKF performs better with the SB structure.
>
> **Scalar-block Number**
>
> | Model                       | ELTO-KF | ELTO-AKF ($k=2$) | ELTO-AKF ($k=5$) | ELTO-AKF ($k=10$) |
> |-----------------------------|---------|------------------|------------------|-------------------|
> | MSE                         | 0.1600  | 0.1043           | 0.1133           | **0.0694**        |
>
> > Impact of Embedding Window Size ($h$) on ELTO-Based Approaches:
>
> As detailed in Appendix C Table 12, we demonstrate the impact of the embedding window size on the nonstationary Lidar trajectory. In this particular nonstationary experiment, smaller window sizes yield lower MSE. However, if the window size becomes too small, it may capture noise rather than meaningful patterns in the time-series data.
>
> **Window Size Ablation**
>
> | Model      | $h=5$       | $h=10$      | $h=20$      |
> |------------|-------------|-------------|-------------|
> | ELTO-KF    | 0.2426      | **0.2412**  | 0.2415      |
> | ELTO-AKF   | **0.1113**  | 0.1296      | 0.1310      |
>
> >Computational Complexity Analysis:
>
> As detailed in Appendix C Table 14, we have included a comprehensive computational complexity analysis. We demonstrate that operator construction scales linearly with the number of training samples and is independent of the block size, $k$. Although a larger $k$ expands the search space—thereby increasing the CMA-ES optimization time ($\mathcal{O}(k^2)$)—it does not affect the operator training or inference times.
>
> **Computational Complexity Analysis of ELTO-AKF**
>
> | Stage                    | ELTO-KF          | ELTO-AKF ($k=2$)  | ELTO-AKF ($k=5$)  | ELTO-AKF ($k=10$) |
> |--------------------------|------------------|-------------------|-------------------|-------------------|
> | Operator Training (s)    | 3.82 ± 0.18      | 3.78 ± 0.02       | 3.80 ± 0.15       | 3.81 ± 0.03       |
> | CMA-ES Optimization (s)  | 137.36 ± 45.27   | 282.91 ± 0.96     | 395.58 ± 1.11     | 507.92 ± 0.71     |
> | Inference (s)            | 1.40 ± 0.01      | 2.778 ± 0.007     | 2.782 ± 0.006     | 2.773 ± 0.008     |

---

> ### Author Response · Authors · 2026-05-01
>
> >Discussion on Memory Factors:
>
> Please refer to Table 10 in the revised manuscript for a sensitivity analysis of the memory factors ($\alpha, \beta$). A comprehensive exploration of alternative strategies for their dynamic adaptation is left for future work.
>
> >Visual Aids for Broader Audience:
>
> We have added a side-by-side comparison (Figure 2 in Section 2.2) between standard Kalman filtering in the original state space and our kernel Kalman filtering in the RKHS to improve clarity for readers who may be less familiar with the Kalman filtering literature.
>
> >Clarification in Table 2:
>
> We have added a clarifying sentence to the main text to better distinguish between these two approaches. The “Pure Adaptation” version requires no dataset-specific CMA-ES optimization and serves as a highly efficient baseline for purely online state estimation. Conversely, the “Full Optimization” regime fine-tunes the hyperparameters of the ELTO-AKF with $\alpha = \beta = 1$ using CMA-ES for a specific dataset.
>
> >Regarding the Broader Impact Statement:
>
> We thank the reviewer for highlighting the dual-use potential of tracking algorithms. Accordingly, we have added a Broader Impact Statement at the end of the revised manuscript (Section 5, Conclusion) to acknowledge the potential repurposing of our robust state estimation technology for surveillance applications.

---

### Review · Reviewer_nd3a · 2026-04-18

**Summary Of Contributions:**

The paper proposes ELTO-AKF, an extension of ELTO-based Kalman filtering that introduces a structured, learnable noise model to handle non-stationary dynamics. It designs a scalar-block parameterization of noise covariances, enabling tractable optimization while preserving positive-definiteness and temporal structure. The method combines noise learning with dynamic adaptation using residuals. Empirically, it improves state estimation and denoising performance across challenging non-stationary tasks, including downstream PDE discovery.

**Audience:**

Yes

**Audience Explanation:**

Yes. Researchers working on Bayesian filtering, kernel methods, and state-space modeling would likely find the paper relevant, especially those interested in non-stationary dynamics and noise modeling. The combination of RKHS-based filtering with adaptive noise learning is technically aligned with TMLR’s audience. However, interest may be narrower outside these subareas due to the methodological focus and lack of real-world validation.

**Broader Impact Concerns:**

None. The paper is about filtering methodology.

**Claims And Evidence:**

Yes

**Claims Explanation:**

The paper provides empirical evidence across multiple tasks, including synthetic filtering benchmarks and PDEs, and shows consistent improvements over baselines. The experiments cover non-stationary settings and different noise regimes, which supports the main claims. However, all validation is on simulated data, with limited real-world evidence. Overall, the evidence is reasonably clear and convincing, but not fully comprehensive.

**Requested Changes:**

1) Can the authors provide an evaluation of ELTO-AKF on at least one real-world dataset, such as object tracking, real data PDE application, or financial time series, where non-stationarity and noise are present? Additionally, does the noise adaptation remain stable and effective when faced with real measurement artifacts (ie outliers) and model mismatch not captured in simulations? Please elaborate.

2) The method appears to require access to ground truth states to learn the noise parameters via an MSE objective. How realistic is this assumption in practical applications where true latent states are typically unavailable, and can the method be adapted to operate without supervised ground truth, for example via a likelihood-based (see my next question).

3) Related to above, why do the authors adopt a derivative-free optimization approach (CMA-ES) combined with heuristic dynamic updates for learning the noise parameters, instead of using a principled likelihood-based approach such as maximum marginal likelihood (MMLE)? This would also avoid the necessity of ground truth data. Can authors then clarify/attempt at implementing MMLE procedures using eg similar techniques to the ones in the ref:

On Particle Methods for Parameter Estimation in State-Space Models. Nikolas Kantas, Arnaud Doucet, Sumeetpal S. Singh, Jan Maciejowski, Nicolas Chopin, https://arxiv.org/abs/1412.8695

I think authors should definitely replace their heuristic parameter fitting procedure (that requires ground truth) with gradient-based methods summarised above, which would make their method much less heuristic and more principled.

4) Please evaluate how ELTO-AKF scales with state and observation dimension, for example on Lorenz-96 system where the dimension of the model can be chosen as a parameter. Such an experiment would help clarify whether the proposed structured covariance parameterization and optimization remain computationally practical and statistically effective as dimensionality increases. Please span the dimension from low (eg 5-10) to hundreds of thousands for a realistic evaluation.

---

> ### Author Response · Authors · 2026-05-01
>
> We sincerely thank you for your insightful comments. According to your suggestions, we have evaluated our method on a real-world dataset as well as high-dimensional systems (Lorenz-96), and included a discussion regarding the feasibility of the MMLE approach in our framework. Below, we address your recommendations and detail the corresponding revisions we have made.
>
> >Evaluation on a real-world dataset and handling unavailable ground truth:
>
> To clarify, our method does not require access to noiseless ground truth. We demonstrate this using a new real-world dataset (historical Canadian lynx and snowshoe hare populations). Since true latent states are unavailable in practice, we utilized the original noisy observations as proxy validation targets (denoted as $\boldsymbol{Y}^{\textrm{val}}$ in the revised manuscript). We evaluated the filter's performance by first perturbing the input data with varying levels of additional synthetic noise and then optimizing the filter to recover the original observations; the recovery MSE is reported in Table 7 in Section 4.5. This approach follows the same setting used for the PDE discovery task in Section 4.4. The results demonstrate that ELTO-AKF is more robust than ELTO-KF in recovering real-world dynamics under nonstationary noise. This success is attributed to our dynamic adaptation mechanism (Algorithm 2), which captures unexpected innovation spikes to adjust the noise covariances, preventing the filter from over-trusting corrupted sensory inputs.
>
> **Filtering performance on the lynx–hare dataset**
>
> | Noise level ($\epsilon$) | ELTO-KF | ELTO-AKF    |
> |--------------------------|---------|-------------|
> | 1                        | 0.0126  | **0.0120**  |
> | 10                       | 0.0569  | **0.0419**  |
> | 100                      | 1.3995  | **1.2602**  |
>
> We also showed that ELTO-AKF can be optimized to smooth the original noisy observations of the two animal populations, allowing for the discovery of a more accurate system of ODEs that is closer to the reference equations, as shown in Figure 5 and Table 8 of the revised manuscript.
>
> >Regarding the MMLE approach and ground truth:
>
> We thank the reviewer for the suggestion. While particle-based MMLE is a principled framework in classical state-space models, its direct application to our setting is non-trivial: our Bayesian inference operates entirely on distribution embeddings (mean embeddings and covariance operators) within the RKHS and, therefore, bypasses explicit probability density functions. Particle-based MMLE, by contrast, requires evaluating a scalar observation likelihood to compute particle weights. Reintroducing such a likelihood into our framework would require either (i) a pre-image projection from the RKHS back into the observation space—an ill-posed inverse problem—or (ii) reimposing a parametric (e.g., Gaussian) assumption on the observation distribution, which would partially negate the non-parametric benefits that motivate the kernel formulation.
>
> Furthermore, our use of CMA-ES is not purely heuristic: as proven in Appendix B, the shift-invariance of the RBF kernel ensures that minimizing the pre-image MSE shares the same minimizer as minimizing the innovation residual within the RKHS, providing a principled distribution-free optimization target. We have made this point more explicit in the revised manuscript and added a discussion of likelihood-based extensions (including particle-based MMLE) as an important direction for future research in the Limitations section.
>
> >Scaling to high-dimensional Lorenz-96 systems:
>
> Following your suggestion, we have added Lorenz-96 experiments with up to $1000$ variables ($D = 1000$) to provide a high-dimensional stress test within our feasible computational resources (Please refer to Table 4 in Section 4.3.). We observed that, in some high-dimensional and high-noise configurations (specifically $D=500, \sigma_{\text{obs}}=0.1$ and $D=1000, \sigma_{\text{obs}}=0.5$), the unstructured ELTO-KF baseline fails to converge due to a loss of symmetric positive-definiteness (SPD), while our structured parameterization remains numerically stable across all tested configurations.

---

### Author Response · Authors · 2026-05-01
**Overview of Revisions**

Dear Editor and reviewers,

We have revised our manuscript based on your constructive feedback. Please refer to the detailed changes in our comments. Here, we summarize the major changes we made below.

**Clarified data assumptions:** We revised the manuscript to clarify that our method uses standard, noisy validation observations ($\boldsymbol{Y}^{\text{val}}$) and does not require noiseless ground truth.

**Added experiments:** We included the Lorenz-96 system to demonstrate our method’s scalability to high-dimensional problems, and the historical Canadian lynx and snowshoe hare dataset to highlight its application to real-world data where the ground truth is unknown.

**New baselines:** We added comparisons with recent Neural-Aided KFs: SPKF and CKFNet.

**Expanded ablation studies:** We added detailed analyses of computational complexity, kernel selection, block size, and embedding window size.

Best regards,

Authors

---

### Decision · Action_Editor_fvuv · 2026-06-08

**Recommendation:** Accept as is

**Audience:**

Yes

**Audience Explanation:**

Definitely. There is a large community of ML researchers interested in dynamical systems embedded within noisy, non-stationary environments who might be interested in the findings of this paper.

**Claims And Evidence:**

Yes

**Claims Explanation:**

The authors have provided additional experiments that better support some of the claims made before. Experiments in general back up the claims in the paper, and the addition of experiments assessing aspects like high-dimensionality has made the evaluation more robust.